# IFNγ-IL12 axis regulates intercellular crosstalk in metabolic dysfunction-associated steatotic liver disease

Randall H. Friedline [1], Hye Lim Noh[1], Sujin Suk[1,2], Mahaa Albusharif[1], Sezin Dagdeviren [1], Suchaorn Saengnipanthkul [1,3], Bukyung Kim[1,4], Allison M. Kim[1], Lauren H. Kim[1], Lauren A. Tauer [1], Natalie M. Baez Torres[1], Stephanie Choi[1], Bo-Yeon Kim[1,5], Suryateja D. Rao[1], Kaushal Kasina[1], Cheng Sun[6], Benjamin J. Toles[6], Chan Zhou [7], Zixiu Li[7], Vivian M. Benoit[1], Payal R. Patel[1], Doris X. T. Zheng[1], Kunikazu Inashima[1], Annika Beaverson[1], Xiaodi Hu[1], Duy A. Tran[1], Werner Muller [8], Dale L. Greiner[1,9], Alan C. Mullen [6], Ki Won Lee[2,10] & Jason K. Kim [1,2,9,11] ✉

Obesity is a major cause of metabolic dysfunction-associated steatohepatitis (MASH) and is characterized by inflammation and insulin resistance. Interferon-γ (IFNγ) is a pro-inflammatory cytokine elevated in obesity and modulating macrophage functions. Here, we show that male mice with loss of IFNγ signaling in myeloid cells (Lyz-IFNγR2$^{-/-}$) are protected from diet-induced insulin resistance despite fatty liver. Obesity-mediated liver inflammation is also attenuated with reduced interleukin (IL)−12, a cytokine primarily released by macrophages, and IL-12 treatment in vivo causes insulin resistance by impairing hepatic insulin signaling. Following MASH diets, Lyz-IFNγR2$^{-/-}$ mice are rescued from developing liver fibrosis, which is associated with reduced fibroblast growth factor (FGF) 21 levels. These results indicate critical roles for IFNγ signaling in macrophages and their release of IL-12 in modulating obesity-mediated insulin resistance and fatty liver progression to MASH. In this work, we identify the IFNγ-IL12 axis in regulating intercellular crosstalk in the liver and as potential therapeutic targets to treat MASH.

[1]Program in Molecular Medicine, University of Massachusetts Chan Medical School, Worcester, MA, USA. [2]WCU Biomodulation Major, Department of Agricultural Biotechnology, College of Agriculture and Life Sciences, Seoul National University, Seoul, Republic of Korea. [3]Division of Nutrition, Department of Pediatrics, Faculty of Medicine, Khon Kaen University, Khon Kaen, Thailand. [4]Division of Endocrinology and Metabolism, Department of Internal Medicine, Kosin University College of Medicine, Busan, Republic of Korea. [5]Division of Endocrinology and Metabolism, Department of Internal Medicine, Soon-chunhyang University Bucheon Hospital, Soonchunhyang University College of Medicine, Bucheon, Republic of Korea. [6]Division of Gastroenterology, Department of Medicine, University of Massachusetts Chan Medical School, Worcester, MA, USA. [7]Division of Biostatistics and Health Services Research, Department of Population and Quantitative Health Sciences, University of Massachusetts Chan Medical School, Worcester, MA, USA. [8]Division of Infection, Immunity & Respiratory Medicine, School of Biological Sciences, University of Manchester, Manchester, United Kingdom. [9]Diabetes Center of Excellence, University of Massachusetts Chan Medical School, Worcester, MA, USA. [10]XO Center, Advanced Institutes of Convergence Technology, Seoul National University, Suwon, Republic of Korea. [11]Division of Endocrinology, Diabetes, and Metabolism, Department of Medicine, University of Massachusetts Chan Medical School, Worcester, MA, USA. ✉e-mail: jason.kim@umassmed.edu

Metabolic dysfunction-associated steatotic liver disease (MASLD) encompasses fatty liver, where lipid deposition accounts for more than 5-10% of the liver's weight, and metabolic dysfunction-associated steatohepatitis (MASH) with the onset of inflammation and fibrosis, leading to cirrhosis. Metabolic liver disease is an emerging health issue affecting one in four adults in the U.S., 30% of individuals over 60 years old, and 75% of subjects with obesity[1-3]. As a major cause of MASLD, obesity is characterized by chronic low-grade inflammation with macrophages infiltrating metabolic organs and increased pro-inflammatory cytokines that may be involved in the pathogenesis of metabolic liver disease[1,4,5].

The liver is an immunological organ, normally enriched in multiple immune cell types, including macrophages and natural killer (NK) cells[6]. During an innate immune response, activated macrophages and NK cells release a multitude of pro-inflammatory cytokines, such as tumor necrosis factor (TNF) α, interleukin (IL)−1β, and IL-6 that orchestrate the interactions between immune cells and surrounding hepatocytes and hepatic stellate cells (HSCs)[4,6,7]. Recent studies have shown that macrophages from lean animals are in the M2-polarized "alternatively-activated" state, but diet-induced obesity induces a phenotypic switch to the M1-polarized "classically-activated" state, actively releasing pro-inflammatory cytokines[8,9]. Emerging evidence further suggests that liver immune cells are also activated in insulin resistance and during MASH progression and that fatty acids may be involved in this pathogenic process[7,10,11].

Interferons (IFNs) are a key component of a complex signaling network regulating innate and adaptive immunity with critical functions in response to infection, cancer surveillance, and metabolic disorders[12]. Interferon-γ (IFNγ) is the only member of the Type II interferon family signaling through a heterodimeric receptor composed of the IFNγR1 and IFNγR2 subunits[13]. As a primary modulator of the macrophage function, IFNγ promotes a strong M1 macrophage response that correlates with macrophage recruitment to metabolic organs in obesity[14,15]. We and others have previously reported that IFNγ levels are elevated in individuals with obesity and obese animals, and reduced IFNγ levels are associated with improved insulin action and glucose metabolism[14,16-18]. While IFNγ has been shown to affect insulin signaling and lipid metabolism in adipocytes[19-21], the effects of IFNγ on liver metabolism are unresolved.

Pro-inflammatory cytokine IL-12, composed of p35 and p40 subunits, plays a unique immunological role in bridging innate and acquired immunity as IL-12 is primarily secreted by antigen-presenting cells, such as macrophages and dendritic cells, and stimulates the proliferation of Th1 cells and NK cells[22-24]. Serum IL-12 (p40) levels are elevated in adults with obesity, *IL-12* and *IL12Rβ2* expressions are increased in MASH liver, and MASH patients are shown to be high producers of IL-12[25-29]. A recent cross-sectional study has also found that serum IL-12 levels are strongly associated with the severe progression of MASLD[30]. In pancreatic islets, IL-12 induces inflammation, causing β-cell dysfunction and apoptosis via activation of signal transducer and activator of transcription (STAT) 4 signaling[31,32]. While systemic IL-12 administration was shown to activate dendritic cells and promote lipid deposition in the liver[33,34], the metabolic role of IL-12 has not been previously addressed.

Here, we used mice with conditional loss of IFNγ signaling in myeloid cells to explore the hypothesis that macrophage IFNγ signaling regulates obesity-mediated insulin resistance in the liver and MASH. We demonstrate that Lyz-IFNγR2$^{-/-}$ mice are protected from diet-induced insulin resistance despite developing fatty liver, and this is associated with improved insulin signaling in the liver. Obesity-mediated liver inflammation is also attenuated in Lyz-IFNγR2$^{-/-}$ mice with profoundly reduced intrahepatic levels of IL-12, and in vivo treatment with IL-12 causes insulin resistance by impairing hepatic insulin signaling in wild-type (WT) mice. A chronic IL-12 treatment re-establishes hepatic insulin resistance and inflammation in Lyz-

IFNγR2$^{-/-}$ mice. Following MASH diets, Lyz-IFNγR2$^{-/-}$ mice are rescued from developing liver fibrosis, which is associated with increased hepatic expression of forkhead box protein O1 (FoxO1) and reduced fibroblast growth factor (FGF) 21 levels. Analysis of RNA-seq datasets from human liver samples showed increased expression of *IFNγR1*, *IFNγR2*, *IL-12B*, and *FGF21* with progressive MASH. Our findings identify critical roles for the IFNγ-IL12 axis in modulating obesity-mediated insulin resistance, inflammation, and fatty liver progression to MASH.

## Results and discussion

### Loss of IFNγ signaling in myeloid cells protects mice from obesity-mediated insulin resistance in the liver

Obesity induces local inflammation in metabolic organs with activated macrophages and increased pro-inflammatory cytokines that are causally associated with insulin resistance[35-37]. Altering macrophage function may be a therapeutic path in treating insulin resistance and MASLD[9,38]. We have previously shown that increased IFNγ levels in skeletal muscle, liver, and adipose tissue are strongly associated with obesity-mediated insulin resistance in these organs[17,18]. Also, modest weight loss via a low-fat diet (LFD) or exercise reduced tissue IFNγ levels that were associated with improved glucose metabolism, suggesting a role for pro-inflammatory cytokine IFNγ in insulin resistance[18]. Mice deficient in IFNγ were shown to have blunted macrophage response, reduced adipose tissue inflammation, and attenuated insulin resistance after high-fat feeding as measured by insulin tolerance tests[14,39,40]. We also found improved insulin sensitivity in global IFNγ null mice (obtained from the Jackson Laboratory) after high-fat feeding, and this was mostly due to improved whole-body glucose turnover and glucose metabolism in skeletal muscle (Supp. Fig. 1A, B). However, liver glucose metabolism and insulin action were not significantly affected by the systemic loss of IFNγ (Supp. Fig. 1C). The organ-selective effects of IFNγ on insulin sensitivity may be due to the pleiotropic effects of IFNγ. While IFNγ was shown to induce adipose tissue inflammation and insulin resistance in peripheral organs, IFNγ also regulates neuronal connectivity[41-43]. Thus, a systemic loss of IFNγ may affect multiple organs and different cellular processes leading to potentially adaptive response, limiting the interpretation of findings from global IFNγ null mice. Importantly, these results indicate that IFNγ per se may not directly affect hepatic glucose metabolism and insulin action in obesity.

Since IFNγ is a primary modulator of the macrophage function, inducing a potent M1-macrophage response[13,14], we generated mice with conditional ablation of IFNγ signaling in myeloid cells using *IFNγR2*-floxed mice crossed to *lysozyme 2* (*LysM*)-Cre mice (Lyz-IFNγR2$^{-/-}$), kindly donated by Dr. Roger Davis (University of Massachusetts Chan Medical School). Lyz-IFNγR2$^{-/-}$ mice on C57BL/6J background were born at the expected Mendelian ratio without obvious anomalies and showed normal metabolic profiles on a standard chow diet. We examined the effects of diet-induced obesity on insulin action in mice. After 10 weeks of a high-fat diet (HFD; 60% kcal from fat), Lyz-IFNγR2$^{-/-}$ and LysM-Cre$^+$ (WT) mice became similarly obese with a 20-30% increase in body weight and a 4-fold increase in whole-body fat mass compared to respective mice fed an LFD (14% kcal from fat, standard chow diet) (Fig. 1A, B). Whole-body lean mass did not differ between genotypes or diet groups (Supp. Fig. 1D). Basal glucose levels were elevated in WT mice after HFD (~180 mg/dl), indicating the onset of type 2 diabetes phenotypes, but not in Lyz-IFNγR2$^{-/-}$ mice (~145 mg/dl) (Fig. 1C). We performed a standardized 2-hour hyperinsulinemic-euglycemic clamp with radioactive-labeled glucose ([3-$^3$H]glucose and 2-[1−$^{14}$C]deoxy-D-glucose) to measure insulin sensitivity and glucose metabolism in awake mice[44]. As expected, WT mice developed insulin resistance after HFD with a 66% decrease in glucose infusion rate required to maintain euglycemia (~130 mg/dl) during clamps compared to LFD-fed WT mice (Fig. 1D & Supp. Fig. 1E). Basal hepatic

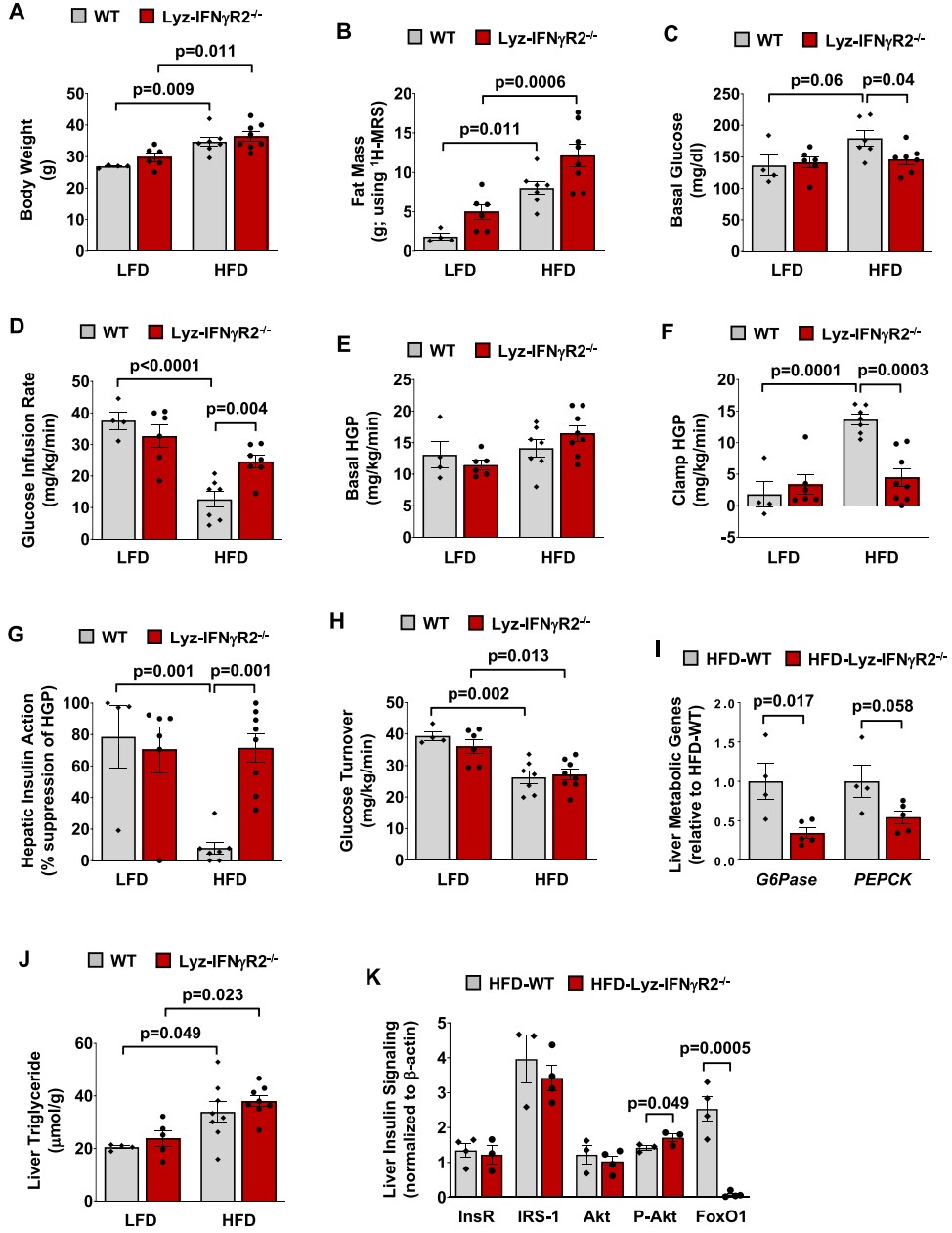

**Fig. 1 | Lyz-IFNγR2⁻/⁻ mice are protected from obesity-mediated insulin resistance in the liver.** Metabolic and molecular experiments were performed in male Lyz-IFNγR2⁻/⁻ and LysM-Cre⁺ (WT) mice at 6 - 7 months of age following 10 weeks of a high-fat diet (HFD; 7 WT and 8 Lyz-IFNγR2⁻/⁻ mice) or a low-fat diet (LFD; 4 WT and 6 Lyz-IFNγR2⁻/⁻ mice) serving as controls. **A** Body weight. **B** Whole-body fat mass, measured using ¹H-magnetic resonance spectroscopy (MRS). **C** Basal plasma glucose levels after overnight fast. **D** Glucose infusion rate required to maintain euglycemia during a standardized 2-hour hyperinsulinemic-euglycemic clamp in awake mice. **E** Basal hepatic glucose production (HGP). **F** Clamp HGP during the insulin-stimulated state. **G** Hepatic insulin action expressed as insulin-mediated percent suppression of basal HGP. **H** Whole-body glucose turnover. **I** RT-qPCR analysis of liver *glucose 6-phosphatase* (*G6Pase*) and *phosphoenolpyruvate carboxykinase* (*PEPCK*) genes in HFD-fed mice (*n* = 4 HFD-WT and 5 HFD-Lyz-IFNγR2⁻/⁻ mice). mRNA levels were

normalized to *HPRT* as a housekeeping gene and are shown relative to HFD-WT mice. **J** Liver triglyceride levels (*n* = 4 LFD-WT, 5 LFD-Lyz-IFNγR2⁻/⁻, 8 HFD-WT, and 8 HFD-Lyz-IFNγR2⁻/⁻ mice). **K** Liver protein expression of insulin receptor (InsR), insulin receptor substrate (IRS)-1, Akt, Ser⁴⁷³-phosphorylation of Akt, and forkhead box protein O1 (FoxO1) using the Jess Multiplexed Western Blot System in HFD-fed mice (*n* = 3 ~ 4 HFD-WT and 3 ~ 4 HFD-Lyz-IFNγR2⁻/⁻ mice). Protein levels were normalized to β-actin as a loading control. Data are presented as mean ± SEM values. The statistical significance of the difference in mean values between Lyz-IFNγR2⁻/⁻ mice and WT mice fed with a HFD or LFD was determined using a one-way analysis of variance (ANOVA) with Tukey's multiple comparison test for post-hoc analysis (Fig. A-H and J). The statistical significance of the difference in mean values between HFD-fed Lyz-IFNγR2⁻/⁻ mice versus HFD-fed WT mice was determined using a two-tailed Student's t-test (Fig. I and K).

glucose production (HGP) was not affected by HFD, but HGP during insulin clamp was increased by almost 8-fold in WT mice after HFD, resulting in a 70% decrease in hepatic insulin action compared to LFD-fed mice (i.e., hepatic insulin resistance) (Fig. 1E–G). Diet-induced insulin resistance in WT mice also involved 33 ~ 47% decreases in whole-body glucose turnover and glycogen plus lipid synthesis after

HFD, reflecting peripheral insulin resistance in these mice (Fig. 1H & Supp. Fig. 1F).

Despite being obese after HFD, Lyz-IFNγR2⁻/⁻ mice were protected from diet-induced insulin resistance with a 2-fold increase in glucose infusion rate compared to WT mice (Fig. 1B & D). This rescue effect was primarily due to reduced HGP during insulin clamp in Lyz-IFNγR2⁻/⁻

mice compared to WT mice on HFD (Fig. 1F). As a result, insulin induced a 77% suppression of HGP (i.e., hepatic insulin action) in Lyz-IFNγR2$^{-/-}$ mice compared to only 8% suppression of HGP in WT mice on HFD (Fig. 1G). Remarkably, such insulin-sensitive liver in obese Lyz-IFNγR2$^{-/-}$ mice after HFD was comparable to liver insulin sensitivity in lean LFD-fed Lyz-IFNγR2$^{-/-}$ mice. Contrary to the effects in the liver, whole-body glucose turnover and glycogen plus lipid synthesis were similarly reduced after HFD in both groups of mice (Fig. 1H & Supp. Fig. 1F). Whole-body glycolysis was not significantly affected by the genotypes or HFD (Supp. Fig. 1G). Thus, these data demonstrate that Lyz-IFNγR2$^{-/-}$ mice were completely rescued from obesity-mediated insulin resistance in the liver.

To determine the molecular mechanism, we performed a real-time quantitative polymerase chain reaction (RT-qPCR) analysis in liver samples collected from HFD-fed mice. We found 45 - 65% decreases in major gluconeogenic genes, *glucose 6-phosphatase* (*G6Pase*) and *phosphoenolpyruvate carboxykinase* (*PEPCK*), in Lyz-IFNγR2$^{-/-}$ mice compared to WT mice (Fig. 1I). This is consistent with reduced HGP and improved liver insulin action in HFD-fed Lyz-IFNγR2$^{-/-}$ mice. Since excess lipid deposition is causally associated with insulin resistance[45,46], we measured liver triglyceride (TG) levels, which were increased by ~60% after HFD in both groups of mice (Fig. 1J). These results indicate that Lyz-IFNγR2$^{-/-}$ mice became obese and developed fatty liver after HFD but were protected from obesity-mediated insulin resistance in the liver.

Next, we performed molecular analysis of insulin signaling using the Jess Multiplexed Western Blot System (ProteinSimple) in liver samples from HFD-fed mice collected at the end of insulin clamps. Liver Ser$^{473}$-phosphorylation of Akt (protein kinase B) was significantly increased by 20% in Lyz-IFNγR2$^{-/-}$ mice compared to WT mice, indicating improved hepatic Akt activity (Fig. 1K). Additionally, FoxO1 is a nuclear transcription factor mediating downstream insulin signaling effects on glucose metabolic genes[47–49], and liver FoxO1 protein levels were dramatically decreased by 97% in Lyz-IFNγR2$^{-/-}$ mice compared to WT mice (Fig. 1K). This is consistent with insulin stimulation of Akt that phosphorylates FoxO1, causing its translocation out of the nucleus and degradation, which results in the suppression of gluconeogenic genes[49–51]. Liver insulin receptor (InsR), insulin receptor substrate (IRS)−1, and total Akt protein levels did not differ between groups (Fig. 1K). These results demonstrate improved liver insulin signaling in HFD-fed Lyz-IFNγR2$^{-/-}$ mice, consistent with decreases in HGP during insulin clamp and gluconeogenic genes in these mice. Our findings further suggest that humoral factors released from myeloid cells upon activation of IFNγ signaling may be a key mediator of liver insulin resistance in obesity.

## Liver inflammation is reduced in HFD-fed Lyz-IFNγR2$^{-/-}$ mice

Obesity is characterized by chronic inflammation with macrophages infiltrating metabolic organs, and increased pro-inflammatory cytokines are causally associated with insulin resistance[35,36,52]. We and others have shown that pro-inflammatory cytokines, such as TNFα and IL-6, impair insulin signaling and glucose metabolism in skeletal muscle and liver[53,54]. Our recent studies found that genetic ablation of 78-kDa glucose-regulated protein or cJun NH$_2$-terminal kinase causes macrophage polarization to an anti-inflammatory M2 state, and these mice are protected from obesity-mediated insulin resistance[9,38].

We performed a multiplexed analysis of cytokines using Luminex in liver samples from HFD-fed Lyz-IFNγR2$^{-/-}$ and WT mice. Despite developing fatty liver after HFD, conditional loss of IFNγ signaling in myeloid cells resulted in a profound reduction in obesity-mediated inflammation, as shown by 20 - 80% decreases in pro-inflammatory cytokines, IL-1β, IL-7, IL-15, IL-17, IFNγ, vascular endothelial growth factor (VEGF), IL-9, KC (murine IL-8 homolog), and monokine-induced by IFNγ (MIG) in the liver of Lyz-IFNγR2$^{-/-}$ mice compared to WT mice (Fig. 2A, B). Notably, IL−12 was dramatically affected by the loss of IFNγ signaling in myeloid cells as liver IL−12 (p40) and IL−12 (p70) levels were decreased by 60 - 90% in Lyz-IFNγR2$^{-/-}$ mice compared to WT

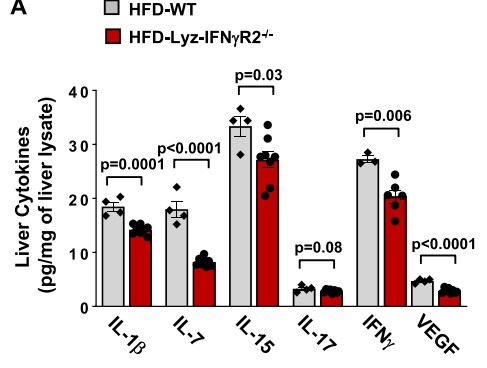

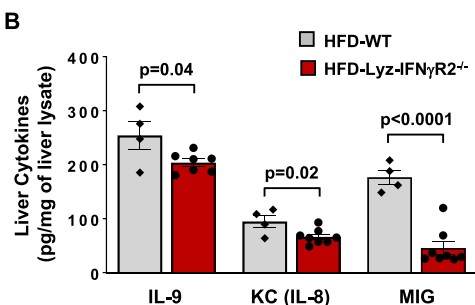

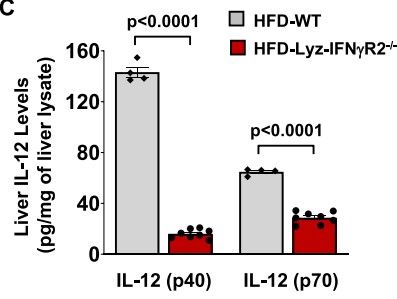

**Fig. 2 | Reduced liver inflammation in HFD-fed Lyz-IFNγR2$^{-/-}$ mice.** Multiplexed-Luminex analysis of liver cytokines was performed in male Lyz-IFNγR2$^{-/-}$ and WT mice at 6 - 7 months of age following 10 weeks of an HFD. **A** IL-1β, IL-7, IL-15, IL-17, IFNγ, and vascular endothelial growth factor (VEGF) levels in the liver (*n* = 3 - 4 HFD-WT and 6 - 8 HFD-Lyz-IFNγR2$^{-/-}$ mice). **B** IL-9, KC (murine IL-8 homolog), and monokine-induced by IFNγ (MIG) levels in the liver (*n* = 4 HFD-WT and 7 - 8 HFD-Lyz-IFNγR2$^{-/-}$ mice). **C** IL-12 (p40) and IL-12 (p70) levels in the liver (*n* = 4 HFD-WT and 8 HFD-Lyz-IFNγR2$^{-/-}$ mice). Data are presented as mean ± SEM values. The statistical significance of the difference in mean values was determined using a two-tailed Student's t-test.

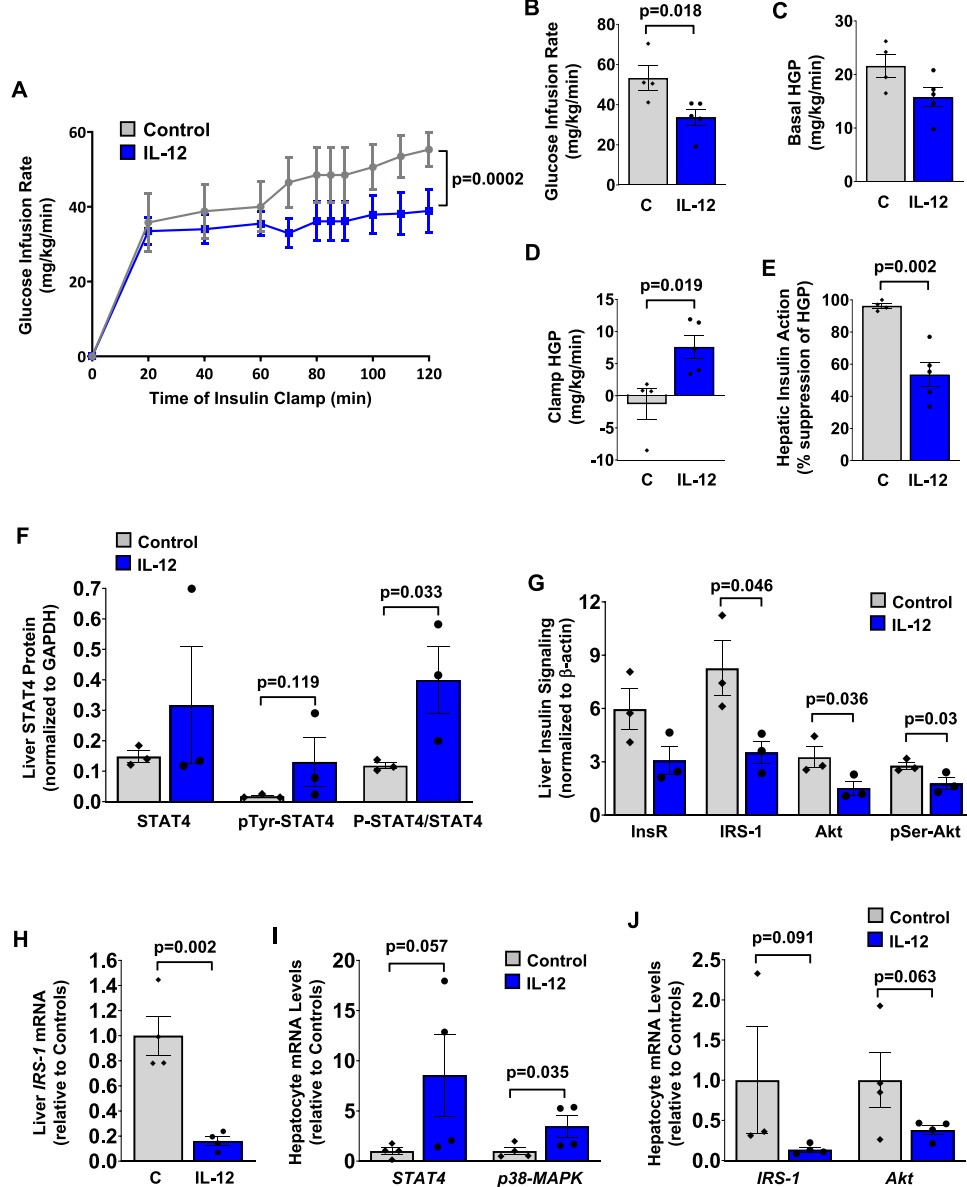

**Fig. 3 | IL-12 treatment in vivo causes liver insulin resistance in wild-type mice.** Mouse recombinant IL-12 (0.25 μg/hour; *n* = 5) or saline (C, Control; *n* = 4) was intravenously administered for 4 hours in male C57BL/6J mice at 4 - 5 months of age, and hyperinsulinemic-euglycemic clamps were performed to measure insulin action and glucose metabolism in awake mice. **A** Time course of glucose infusion rates required to maintain euglycemia (~100 mg/dl) during clamps. **B** Steady-state glucose infusion rates during 90-120 min of clamps. **C** Basal HGP. **D** Clamp HGP during the insulin-stimulated state. **E** Hepatic insulin action expressed as insulin-mediated percent suppression of basal HGP. **F** Liver samples were collected 1 hour after IL-12 injection (1 mg) with saline serving as control, and the Jess Multiplexed Western Blot System was used to measure the liver expression of signal transducer and activator of transcription 4 (STAT4), Tyr[693]-phosphorylation of STAT4, and a ratio of phospho-STAT4 to total STAT4 protein levels (*n* = 3 per group). Protein levels were normalized to GAPDH as a loading control. **G** Insulin signaling was analyzed using liver samples collected at the end of the insulin clamp following 4 hours of IL-12 (*n* = 3) or saline (*n* = 3) infusion. Liver protein levels of insulin receptor (InsR), IRS-1, Akt, and Ser[473]-phosphorylation of Akt were determined using the Jess Western Blot System (*n* = 3 per group). Protein levels were normalized to β-actin as a loading control. **H** RT-qPCR analysis of liver *IRS-1* mRNA levels normalized to *HPRT* and shown relative to controls (*n* = 4 per group). **I, J** Primary hepatocytes were isolated from male C57BL/6J mice at 4 - 5 months of age following a 4-hour intravenous infusion of IL-12 (0.25 μg/hour; *n* = 4) or saline (control; *n* = 4) for RT-qPCR analysis of *STAT4*, *p38-mitogen-activated protein kinase* (*MAPK*), *IRS-1*, and *Akt*. mRNA levels were normalized to *HPRT* as a housekeeping gene and shown relative to controls. Data are presented as mean ± SEM values. The statistical significance of the difference in mean values was determined using a two-tailed Student's t-test or a one-tailed Student's t-test for Fig. **F**, **I**, and **J**.

mice (Fig. 2C). Since IL-12 is released by macrophages in response to IFNγ signaling, we next explored a possible role for IL-12 as a humoral factor regulating insulin action and glucose metabolism in the liver.

## IL-12 treatment in vivo causes insulin resistance in the liver

A heterodimeric ligand IL-12 plays a key immunological role in bridging innate and acquired immunity as IL-12 is primarily secreted by antigen-presenting cells, such as macrophages, and serum IL-12 (p40) levels are elevated in individuals with obesity[22–26]. To examine the effects of IL-12 on glucose metabolism, mouse recombinant IL-12 (0.25 μg/hour) or saline (control) was intravenously administered for 4 hours in male WT mice (C57BL/6J), followed by a standardized hyperinsulinemic-euglycemic clamp in awake mice. Mice were randomized into 2 experimental groups and showed similar body weights

and whole-body fat/lean mass (Supp. Fig. 1H). Basal plasma glucose levels did not differ between groups, and glucose levels were matched during insulin clamps (Supp. Fig. 1I). IL−12 treatment in vivo caused a 30% decrease in glucose infusion rates during 90−120 minutes of the clamps (steady-state) compared to controls, indicating blunted insulin action in IL12-treated mice (Fig. 3A, B). To our knowledge, this is the first observation of IL−12 causing insulin resistance in vivo, and this was due to increased HGP during insulin clamp in IL12-treated mice compared to controls with no effects on basal HGP (Fig. 3C, D). As a result, insulin induced a 54% suppression of HGP following IL−12 treatment compared to 96% suppression of HGP in controls, indicating hepatic insulin resistance in IL12-treated mice (Fig. 3E). Whole-body glucose turnover was not significantly affected by IL−12 treatment (Supp. Fig. 1J).

Upon binding to a heterodimeric receptor, IL−12 mediates intracellular signaling by Janus tyrosine kinases phosphorylating IL12Rβ2 and activation of STAT3 and STAT4 via Tyr/Ser-phosphorylation[55–57]. We examined the effects of IL-12 on molecular signaling pathways by collecting liver samples 1 hour after IL-12 injection (1 mg) with saline serving as controls and using the Jess Multiplexed Western Blot System. Our liver protein analysis revealed that IL-12 induced more than a 3-fold increase in the ratio of Tyr[693]-phosphorylation of STAT4 to total STAT4 protein levels compared to controls (Fig. 3F). IL-12 injection did not significantly affect total STAT4 protein levels in the liver (Fig. 3F). These data are consistent with an acute effect of IL-12 to potently induce STAT4 phosphorylation and a chronic effect of IL-12 to increase STAT4 protein expression in the liver[56]. To determine the molecular mechanism by which IL-12 causes hepatic insulin resistance, we examined insulin signaling in liver samples collected at the end of the insulin clamp following 4 hours of IL-12 infusion (0.25 µg/hour). Our analysis using the Jess Multiplexed Western Blot System found that IL-12 treatment caused 36 ~ 57% decreases in liver IRS-1, Akt, and Ser[473]-phosphorylation of Akt protein levels compared to controls (Fig. 3G). RT-qPCR analysis further showed an 84% decrease in liver *IRS-1* mRNA levels after IL-12 treatment (Fig. 3H). Insulin stimulation of IRS-1 and Akt is a principal pathway by which insulin regulates glucose metabolism in the liver[58], and these results are consistent with IL-12 effects to increase HGP and impair hepatic insulin action. Since IL-12 did not affect InsR protein levels in the liver (Fig. 3G), these data suggest that IL12-mediated insulin resistance involves defects in hepatic insulin signaling downstream of insulin receptors.

Since the liver is a heterogeneous organ consisting of multiple cell types, we next cultured primary hepatocytes from WT mice after 4 hours of IL-12 (0.25 µg/hour) or saline (control) infusion. RT-qPCR analysis found that IL-12 increased hepatocyte expression of *STAT4* by more than 8-fold over controls (Fig. 3I). Previous studies have shown IL-12 activation of mitogen-activated protein kinase (MAPK) kinases 3 and 6 and Thr/Tyr-phosphorylation of p38-MAPK, a known inhibitor of insulin signaling in adipocytes[59–61]. In that regard, IL-12 also increased hepatocyte expression of *p38-MAPK* by more than 3-fold over controls (Fig. 3I). Consistent with the liver protein data, hepatocyte mRNA levels of *IRS-1* and *Akt* were decreased by 57 to 86% following IL-12 treatment (Fig. 3J). Taken together, these results indicate that IL-12 is a pro-inflammatory cytokine causing insulin resistance in the liver, and the underlying mechanism involves IL12-mediated downregulation of hepatic insulin signaling possibly via activation of p38-MAPK. Our findings further support the notion that macrophage IFNγ signaling, through the release of IL-12, modulates obesity-mediated insulin resistance in the liver.

## Conditional loss of IFNγ signaling in myeloid cells rescues mice from MASH

While fatty liver and MASH share common risk factors, such as obesity and type 2 diabetes, recent studies indicate that only 20% of individuals with fatty liver progress to develop MASH, suggesting an

important role for liver inflammation in MASH pathogenesis[3]. Since Lyz-IFNγR2$^{-/-}$ mice are protected from obesity-mediated inflammation and "cytokine storm" in the liver, we next explored whether such protection affects MASH development in these mice. Male Lyz-IFNγR2$^{-/-}$ and WT mice were fed with a methionine-choline deficient (MCD) diet, and liver fibrosis and steatosis were noninvasively measured using a state-of-the-art Vega Wide-field Ultrasound Imaging System (SonoVol) (Fig. 4A showing representative 3D ultrasound images). After 4 weeks on the MCD diet, WT mice developed severe liver fibrosis, but Lyz-IFNγR2$^{-/-}$ mice showed a significant attenuation of liver fibrosis, as indicated by a 20% decrease in liver stiffness (Fig. 4B). This remarkable rescue phenotype was also visually evident in liver images of WT mice showing regions of high intensity (red-color nearing 20 kPa) whereas Lyz-IFNγR2$^{-/-}$ mice predominantly showed regions of low intensity (blue-color nearing 0 kPa) following MCD diet (Fig. 4C).

Since the MCD diet causes weight loss, cachexia, and hepatocyte death due to oxidative stress[62], we next examined the effects of a choline-deficient L-amino acid HFD (CDAHFD) in Lyz-IFNγR2$^{-/-}$ and WT mice. Both groups of mice rapidly developed fatty liver with steatosis increasing by 2-fold after 4 weeks of CDAHFD (Fig. 4D). Despite early onset steatosis, liver fibrosis gradually increased in WT mice, but this increase was noticeably blunted in Lyz-IFNγR2$^{-/-}$ mice (Fig. 4E). After 20 weeks of CDAHFD, Lyz-IFNγR2$^{-/-}$ mice showed a 20% decrease in liver fibrosis compared to WT mice, and this is further reflected in representative liver 3D images showing shear wave elasticity (SWE) and liver histology images with Masson's trichrome stain (Fig. 4E–G). This protection from CDAHFD-induced MASH phenotypes in Lyz-IFNγR2$^{-/-}$ mice is consistent with the results from the MCD diet.

While CDAHFD is widely used to model MASH phenotypes, we found that mice did not gain body weight during 20 weeks of CDAHFD (Supp. Fig. 2A), which was previously observed[63,64]. In that regard, recent studies have shown that mice fed with the Gubra Amylin NASH (GAN) diet develop obesity and liver lesions with morphological characteristics closely resembling human MASH[65–67]. During 20 weeks on the GAN diet, Lyz-IFNγR2$^{-/-}$ and WT mice became obese with a 30 ~ 40% increase in body weight and developed fatty liver with a gradual increase in liver steatosis (Fig. 5A, B). WT mice showed a progressive increase in liver fibrosis on the GAN diet, but Lyz-IFNγR2$^{-/-}$ mice were completely rescued from developing liver fibrosis during 20 weeks on the GAN diet (Fig. 5C). This is also evident in representative liver images showing shear wave elasticity from mice after 20 weeks on the GAN diet (Fig. 5D).

Next, we performed a detailed histological analysis of liver samples collected from Lyz-IFNγR2$^{-/-}$ (KO) and WT mice following the GAN diet. This involved a comprehensive pathology evaluation of liver sections processed for hematoxylin & eosin (H&E) and Masson's Trichrome stain by a board-certified pathologist (Applied Pathology Systems) for the grading of inflammatory infiltrate, steatosis, hepatocellular injury, bile duct injury, and fibrosis. We found that there was an across-the-board decrease in the histology scores for inflammatory infiltrate, hepatocellular injury, and fibrosis in Lyz-IFNγR2$^{-/-}$ mice compared to WT mice after 20 weeks on the GAN diet (Supp. Table 1), and this is consistent with liver fibrosis data obtained from the ultrasound imaging (Fig. 5C). In contrast, macro-vesicular and micro-vesicular scores for steatosis did not differ between both groups of GAN diet-fed mice (Supp. Table 1), and this is also consistent with liver steatosis data obtained from the ultrasound imaging (Fig. 5B). As a result, the total pathology scores were significantly reduced in Lyz-IFNγR2$^{-/-}$ mice compared to WT mice after 20 weeks on the GAN diet (Supp. Table 1). Additionally, we evaluated MASLD Activity Score (MAS), originally validated as a scoring system for human MASH[68,69], and found a strong trend for less lobular inflammation and ballooning but similar steatosis in Lyz-IFNγR2$^{-/-}$ mice compared to WT mice after the GAN diet, resulting in a modestly lower overall MAS in

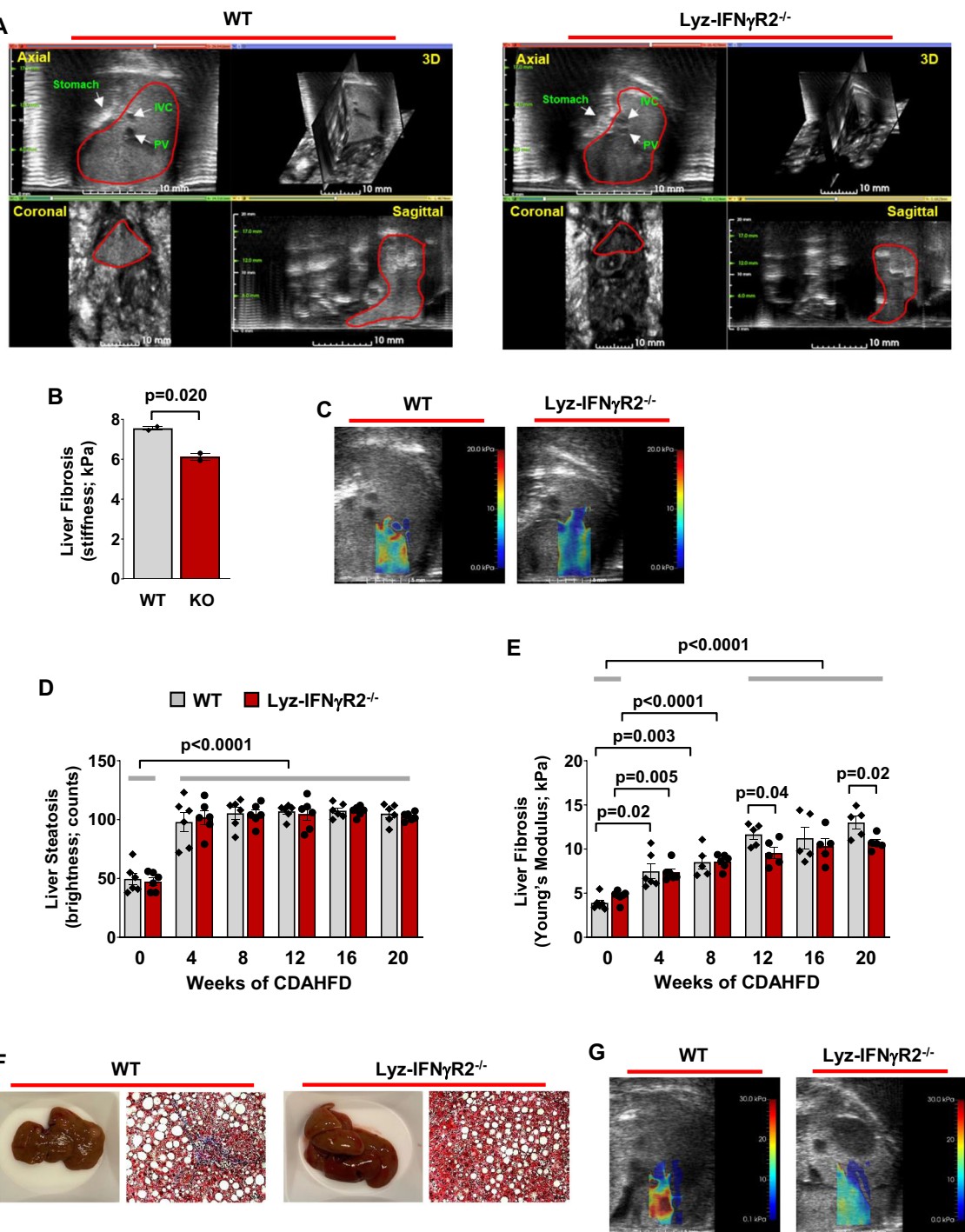

**Fig. 4 | Lyz-IFNγR2⁻/⁻ mice are rescued from fatty liver progression to MASH.**
Noninvasive ultrasound imaging and histological analyses were performed to assess MASH phenotypes in male Lyz-IFNγR2⁻/⁻ and WT mice after 4 weeks of a methionine-choline deficient (MCD) diet starting at 6 - 7 months of age ($n = 3$ per genotype). **A** Representative ultrasound images showing axial, 3D, coronal, and sagittal views using Vega Wide-field Ultrasound Imaging System in mice after 4 weeks of MCD diet. **B** Liver fibrosis measured as stiffness using Young's Modulus in MCD diet-fed mice. **C** Representative shear wave elastography (SWE) images showing shear wave elasticity in regions of high intensity (red-color nearing 20 kPa) and regions of low intensity (blue-color nearing 0 kPa) in MCD diet-fed mice. Additional cohorts of male Lyz-IFNγR2⁻/⁻ and WT mice were used to assess MASH phenotypes during 20 weeks of a choline-deficient L-amino acid HFD (CDAHFD) starting at ~8 months of age ($n = 6$ per genotype). **D** Liver steatosis as shown in brightness during 20 weeks of CDAHFD in mice. The mean values of liver steatosis

in Lyz-IFNγR2⁻/⁻ and WT mice at 4, 8, 12, 16, and 20 weeks of CDAHFD versus 0 week are statistically significant ($p < 0.0001$). **E** Liver fibrosis as stiffness using Young's Modulus during 20 weeks of CDAHFD in mice. The mean values of liver fibrosis in Lyz-IFNγR2⁻/⁻ and WT mice at 12, 16, and 20 weeks of CDAHFD versus 0 week are statistically significant ($p < 0.0001$). **F** Representative liver photos and histology images with Masson's trichrome stain (collagen stained in blue) from CDAHFD-fed mice. **G** Representative SWE images showing shear wave elasticity in regions of high and low intensity in CDAHFD-fed mice. Data are presented as mean ± SEM values. The statistical significance of the difference in mean values between Lyz-IFNγR2⁻/⁻ mice versus WT mice at each time point (weeks of CDAHFD) was determined using a two-tailed Student's t-test. The statistical significance of the difference in mean values between 4, 8, 12, 16, and 20 weeks of CDAHFD versus 0 week was determined using a one-way analysis of variance (ANOVA) with Tukey's multiple comparison test for post-hoc analysis.

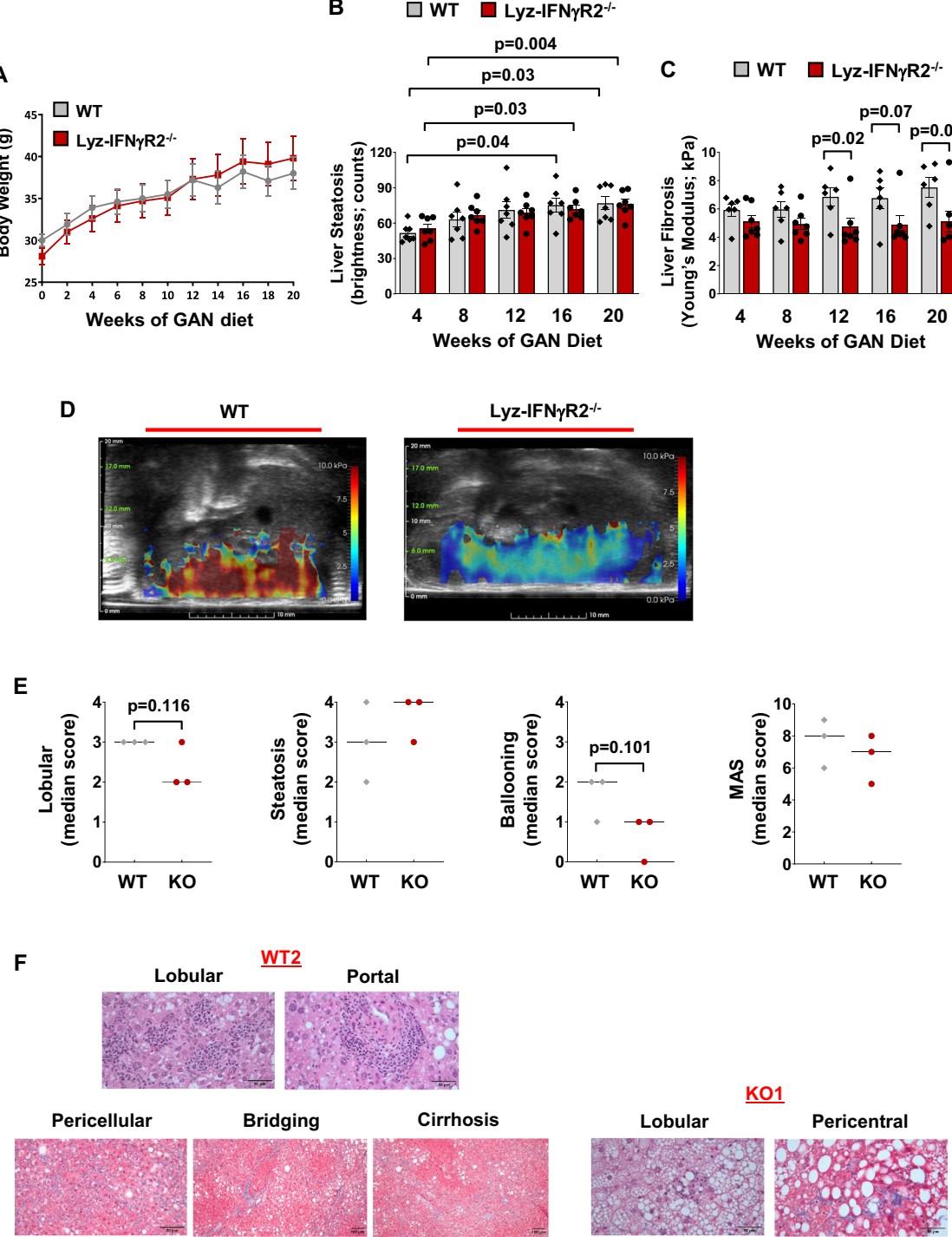

**Fig. 5 | Obese Lyz-IFNγR2⁻/⁻ mice are protected from liver fibrosis but not steatosis after the GAN diet.** Noninvasive ultrasound imaging and histological analyses were performed to assess MASH phenotypes in male Lyz-IFNγR2⁻/⁻ and WT mice during 20 weeks of the Gubra Amylin NASH (GAN) diet starting at 6 months of age (*n* = 7 per genotype). **A** Body weight changes during 20 weeks of the GAN diet. **B** Liver steatosis as shown in brightness during 20 weeks of the GAN diet. **C** Liver fibrosis as stiffness using Young's Modulus during 20 weeks of the GAN diet (*n* = 6 WT and 7 Lyz-IFNγR2⁻/⁻ mice). **D** Representative SWE images showing shear wave elasticity in regions of high and low intensity in GAN diet-fed mice. **E** Liver samples collected after 20 weeks of the GAN diet were processed for hematoxylin & eosin (H&E) and Masson's Trichrome stain, and histology slides were evaluated by a board-certified pathologist (Applied Pathology Systems). MASLD activity score

(MAS) was calculated for individual Lyz-IFNγR2⁻/⁻ and WT mice (*n* = 3 per genotype) by combining the histology scoring for lobular inflammation, macro-vesicular steatosis, and ballooning. **F** Representative histology images showing lobular and portal inflammation, and pericellular, bridging, and cirrhotic nodular fibrosis in WT2 mouse. Histology images showing lobular inflammation and pericentral fibrosis in KO1 mouse. Data are presented as mean ± SEM values. The statistical significance of the difference in mean values between Lyz-IFNγR2⁻/⁻ mice versus WT mice at each time point (weeks of GAN diet) was determined using a two-tailed Student's t-test. The statistical significance of the difference in mean values between 8, 12, 16, and 20 weeks of GAN diet versus 4 weeks was determined using a one-way analysis of variance (ANOVA) with Tukey's multiple comparison test for post-hoc analysis.

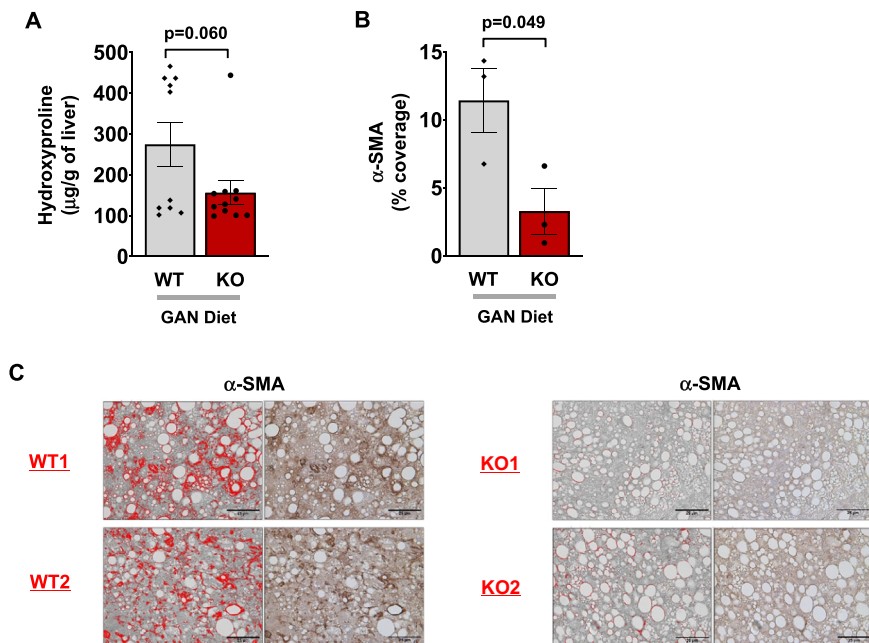

**Fig. 6 | Reduced biomarkers for liver fibrosis and HSC activation in Lyz-IFNγR2$^{-/-}$ mice following the GAN diet.** Liver hydroxyproline and α-smooth muscle actin (α-SMA) contents were measured as reliable markers of fibrosis and activation of hepatic stellate cells (HSC), respectively. **A** Hydroxyproline content in liver samples collected from WT mice ($n = 10$) and Lyz-IFNγR2$^{-/-}$ mice (KO; $n = 11$) after 20 weeks of the GAN diet. **B** Percent coverage of α-SMA stain in liver sections from WT and KO mice after the GAN diet ($n = 3$ per genotype). **C** Histology images showing α-SMA stain for WT1, WT2, KO1, and KO2 mice with stain areas highlighted in red. Data are presented as mean ± SEM values. The statistical significance of the difference in mean values was determined using a two-tailed Student's t-test.

Lyz-IFNγR2$^{-/-}$ mice (Fig. 5E). The representative histology images from WT and Lyz-IFNγR2$^{-/-}$ (KO) mice are shown in Fig. 5F. Taken together, these histology results support the ultrasound imaging data indicating that Lyz-IFNγR2$^{-/-}$ mice are protected from liver fibrosis but not steatosis following the MASH diets.

Moreover, we measured liver hydroxyproline content as the gold standard for fibrosis assessment and found 43% and 33% decreases in liver hydroxyproline levels in Lyz-IFNγR2$^{-/-}$ mice compared to WT mice after 20 weeks on the GAN diet and CDAHFD, respectively (Fig. 6A and Supp. Fig. 2B). The α-smooth muscle actin (α−SMA), a protein encoded by the gene *ACTA2* in humans, is an actin isoform that increases in expression during HSC activation and therefore serves as a reliable marker of activated HSCs during early stages of liver fibrogenesis[70,71]. Liver sections from GAN-diet fed mice were processed for α−SMA stain and showed a 71% decrease in percent coverage of α−SMA stain in Lyz-IFNγR2$^{-/-}$ mice compared to WT mice, indicating attenuated HSC activation in Lyz-IFNγR2$^{-/-}$ mice (Fig. 6B). The representative histology images of α−SMA stain from 2 WT and 2 KO livers are shown in Fig. 6C. Overall, these results indicate that IFNγ-signaling in myeloid cells plays a critical role in obesity-mediated liver inflammation and fatty liver progression to MASH.

### IL-12 modulates hepatocyte FoxO1 and release of FGF21
Circulating levels of IL-12 are increased in individuals with obesity and obese animals, and MASH patients are shown to be high producers of IL-12[25–27]. We found that IL-12 causes insulin resistance by decreasing IRS1/Akt-associated insulin signaling in the liver. Paradoxically, we observed that IL-12 caused a 2-fold increase in IRS-2 protein levels and an 82% increase in *IRS-2* mRNA levels in the liver (Fig. 7A, B). Previous studies have shown a reciprocal regulation between IRS-1 and IRS-2 in the liver, and increased IRS-2 expression might be a compensatory response to IL12-mediated downregulation of IRS-1 activity in the liver[72,73]. Liver mRNA levels of *protein-tyrosine phosphatase SHP1* (*Ptpn6*), which modulates insulin signaling activity, were also reduced by 38% in IL12-treated mice (Fig. 7B)[74,75].

Moreover, liver expression of the FoxO1 transcription factor was decreased by 20% (mRNA levels) and by 41% (protein levels) in IL12-treated mice compared to controls (Fig. 7B, C). This may be due to IL-12 activation of p38-MAPK, which was shown to regulate the transcriptional activity of FoxO1 via phosphorylation[76]. Additionally, Deng et al. have found that pro-inflammatory cytokines, IL-2, IL-12, and IL-15, induce phosphorylation of FoxO1, leading to its inactivation in NK cells[77]. IL-12 was also shown to promote the highest level of FoxO1 phosphorylation after 2 hours of treatment[77]. Thus, hepatocyte transcription factor FoxO1 may be competitively regulated by insulin and pro-inflammatory cytokines, possibly leading to different pathogenic outcomes for the liver[49,78]. RT-qPCR analysis of primary hepatocytes isolated from WT mice following IL-12 treatment further showed a 9-fold increase in *IRS-2* mRNA levels and a 41% decrease in *FoxO1* mRNA levels compared to controls (Fig. 7D, E), supporting the IL-12 regulation of hepatocyte IRS2-FoxO1 signaling pathway.

Hepatocyte transcription factor FoxO1 was recently shown to be a potent negative regulator of FGF21 secretion from the liver[48,79,80]. In that regard, we found a 20-fold increase in hepatocyte *FGF21* mRNA levels and a 2.5-fold increase in circulating FGF21 protein levels in IL12-treated mice compared to controls (Fig. 7F, G). These data are consistent with IL12-mediated downregulation of FoxO1, subsequently increasing FGF21 secretion from the liver. This is significant because FGFs play a major role in tissue repair by stimulating cell proliferation and providing paracrine and endocrine signals in liver development and pathogenesis[81–83]. In fact, FGF21, an endocrine factor secreted by hepatocytes, regulates HSCs by signaling through FGFR1C with β-klotho acting as the coreceptor, and selective inhibition of FGFR1 has been shown to reduce HSC activation and liver collagen deposition in rats[81,82,84–86]. FGF21 levels are increased in individuals with MASH and strongly associated with MASH severity in patients with obesity and type 2 diabetes[87,88]. Nonetheless, the role of FGF21 in MASH remains controversial as exogenous FGF21 treatment reduced hepatic fibrosis in obese mice, and a phase 2 clinical trial found improved fibrosis biomarkers in FGF21-treated subjects with obesity[84–90]. However, the

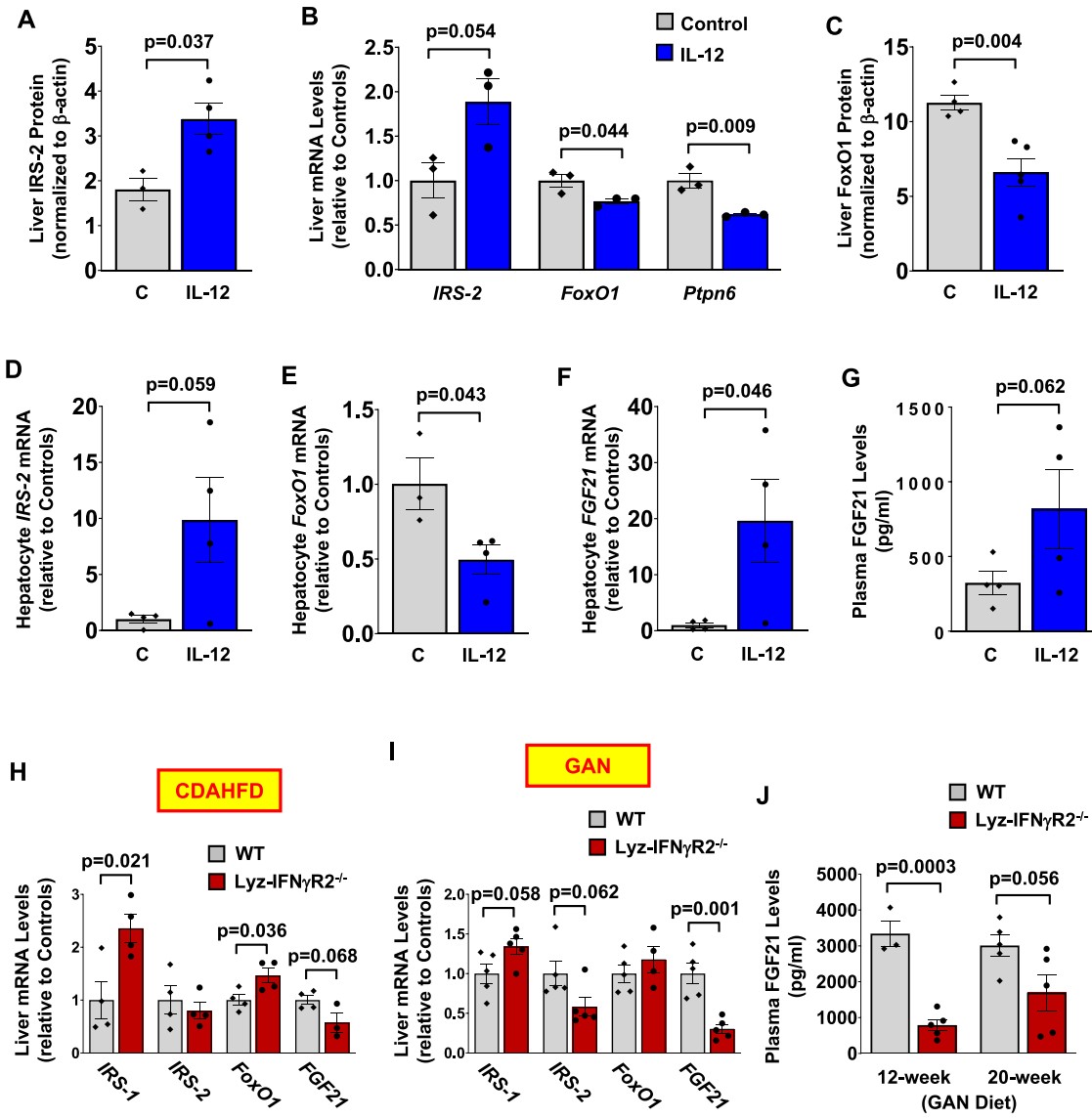

**Fig. 7 | IL-12 regulates the IRS/FoxO1/FGF21 pathway in the liver.** Liver and plasma samples and primary hepatocytes were isolated from male WT mice (C57BL/6J) treated with IL-12 (0.25 µg/hour) or saline (C, Control) for molecular analyses. **A** Liver IRS-2 protein levels measured using Jess Multiplexed Western Blot System (*n* = 3 Controls and 4 IL12-treated mice). **B** RT-qPCR analysis of *IRS-2, FoxO1*, and *protein-tyrosine phosphatase SHP1* (*Ptpn6*) mRNA levels in the liver. (*n* = 3 per group) **C** Liver FoxO1 protein levels measured using Jess Western Blot System. (*n* = 4 Controls and 5 IL12-treated mice). **D-F** Primary hepatocytes were isolated from male WT mice at 4 - 5 months of age following a 4-hour intravenous infusion of IL-12 (0.25 µg/hour; *n* = 4) or saline (Control; *n* = 4) for RT-qPCR analysis of *IRS-2, FoxO1*, and *fibroblast growth factor 21* (*FGF21*). mRNA levels were normalized to *HPRT* and shown relative to controls. **G** Plasma FGF21 levels using ELISA. (*n* = 4 per group). **H & I** Liver samples were collected from male Lyz-IFNγR2⁻/⁻ and WT mice after 20 weeks of CDAHFD or GAN diet for RT-qPCR analysis of *IRS-1, IRS-2, FoxO1*, and *FGF21* mRNA levels in the liver. (*n* = 4 per genotype for CDAHFD-fed mice and *n* = 5 per genotype for GAN diet-fed mice). **J** Plasma FGF21 levels were measured using ELISA in male Lyz-IFNγR2⁻/⁻ and WT mice after 12 and 20 weeks of the GAN diet. (*n* = 3 - 5 WT and 5 Lyz-IFNγR2⁻/⁻ mice) All protein levels were normalized to β-actin as a loading control. All mRNA levels were normalized to *HPRT* as a housekeeping gene and shown relative to controls. Data are presented as mean ± SEM values. The statistical significance of the difference in mean values was determined using a two-tailed Student's t-test.

beneficial effects of FGF21 observed in liver pathology may be secondary to the action of FGF21 on energy expenditure and systemic metabolism[85,90–93].

### Altered IRS/FoxO1/FGF21 pathway in the liver of Lyz-IFNγR2⁻/⁻ mice after MASH diets

To determine whether the liver IRS/FoxO1/FGF21 pathway is involved in the protective effects of IFNγ signaling-deficient macrophages on MASH phenotypes, we examined the liver expression of metabolic genes in Lyz-IFNγR2⁻/⁻ mice that were rescued from liver fibrosis after MASH diets. Liver *IRS-1* mRNA levels were increased by 2.4-fold and 34% in Lyz-IFNγR2⁻/⁻ mice compared to WT mice after CDAHFD and GAN diet, respectively (Fig. 7H, I). In contrast, liver *IRS-2* mRNA levels were decreased by 18 - 36% in Lyz-IFNγR2⁻/⁻ mice compared to WT mice following MASH diets (Fig. 7H, I). This reciprocal change in *IRS-1* and *IRS-2* gene expression is consistent with the IL-12 effects on liver IRS signaling. Moreover, liver *FoxO1* mRNA levels were increased by 18 - 46% in Lyz-IFNγR2⁻/⁻ mice compared to WT mice after MASH diets, and this was associated with 43 - 70% decreases in liver *FGF21* mRNA levels in Lyz-IFNγR2⁻/⁻ mice after MASH diets (Fig. 7H, I). Circulating levels of FGF21 were markedly reduced by 76% and 44% in Lyz-IFNγR2⁻/⁻ mice compared to WT mice after 12 and 20 weeks on the GAN diet, respectively (Fig. 7J). These changes in liver expression of IRS-FoxO1-FGF21 in Lyz-IFNγR2⁻/⁻ mice are in direct contrast to the effects of IL-12 on liver metabolic genes. Thus, these results support the notion that mice with IFNγ signaling-deficient macrophages are protected from

MASH due to blunted IL-12 effects on hepatocyte IRS/FoxO1 signaling and reduced FGF21 release and action on liver fibrogenesis.

## IL-12 treatment re-establishes liver insulin resistance, inflammation, and MASH phenotypes in Lyz-IFNγR2$^{-/-}$ mice

The causal role of IL-12 as a macrophage-derived cytokine downstream of IFNγ signaling to modulate liver insulin resistance was examined by a chronic IL-12 treatment in HFD-fed Lyz-IFNγR2$^{-/-}$ mice that were previously shown to be protected from diet-induced insulin resistance in the liver. Male Lyz-IFNγR2$^{-/-}$ mice (KO) and WT mice were fed an HFD for 12 weeks, and a mouse recombinant IL-12 was chronically administered via osmotic pumps (1.2 ng/day/g body weight; Alzet) during the last 2 weeks of HFD. Body weights and whole-body fat mass tended to be lower in HFD-fed Lyz-IFNγR2$^{-/-}$ mice compared to HFD-fed WT mice following IL-12 treatment although this difference did not reach a statistical significance (Fig. 8A, B). A chronic IL-12 treatment restored type 2 diabetes phenotypes (i.e., hyperglycemia and insulin resistance) in HFD-fed Lyz-IFNγR2$^{-/-}$ mice, showing similar levels of basal glycemia and glucose infusion rates during hyperinsulinemic-euglycemic clamps to the HFD-fed WT mice (Fig. 8C, D, Supp. Fig. 1K). This was primarily due to the deleterious effects of IL-12 in re-establishing hepatic insulin resistance as HGP was minimally suppressed by insulin in Lyz-IFNγR2$^{-/-}$ mice after IL-12 treatment and to a level comparable to the WT mice (Fig. 8E). As a result, hepatic insulin action was measured at ~30% for both groups of HFD-fed mice following IL-12 treatment (Fig. 8F), and this was profoundly lower than the 77% hepatic insulin action previously measured in HFD-fed Lyz-IFNγR2$^{-/-}$ mice (Fig. 1G). Whole-body glucose turnover, reflecting peripheral insulin action, tended to be higher in Lyz-IFNγR2$^{-/-}$ mice following IL-12 treatment (p = 0.086), which may be attributed to lower obesity in these mice (Fig. 8G). Whole-body glycolysis and glycogen/lipid synthesis (GS) did not differ between groups (Supp. Fig. 1L). Thus, these results indicate that Lyz-IFNγR2$^{-/-}$ mice lost their protection from diet-induced insulin resistance in the liver following a chronic IL-12 treatment.

Next, male Lyz-IFNγR2$^{-/-}$ mice (KO) and WT mice were fed the GAN diet for 8 weeks, and IL-12 was chronically administered via osmotic pumps (1.2 ng/day/g body weight) during the last 2 weeks of the GAN diet. Using Vega Wide-field Ultrasound Imaging System, we found that Lyz-IFNγR2$^{-/-}$ and WT mice developed a comparable degree of liver steatosis and fibrosis following the GAN diet and IL-12 treatment, indicating that IL-12 re-established fatty liver progression to liver fibrosis in Lyz-IFNγR2$^{-/-}$ mice (Fig. 8H, I). Representative SWE images are shown in Fig. 8J. Lastly, we performed a multiplexed analysis of cytokines using Luminex in liver samples collected at the end of the GAN diet and IL-12 treatment. Pro-inflammatory cytokines/chemokines, including IL-1β, IL-6, IL-7, KC, IL-9, IL-12 (p70), IL-15, IL-17, IFNγ, VEGF, MIG, and monocyte chemoattractant protein-1 (MCP-1), were similarly elevated in the livers of Lyz-IFNγR2$^{-/-}$ and WT mice following the GAN diet and IL-12 treatment (Fig. 8K, L). Taken together, a chronic IL-12 treatment re-established diet-induced insulin resistance in the liver and the effects of the GAN diet to induce liver inflammation and fibrosis in Lyz-IFNγR2$^{-/-}$ mice, implicating a pivotal role for IL-12 in metabolic liver disease.

Obesity is a major cause of MASLD with inflammation unfolding in its pathogenesis, but the underlying mechanism remains unclear[1,2]. While obesity prevalence continues to climb globally, the U.S. Food and Drug Administration just recently approved Rezdiffra (resmetirom) as the first medication for the treatment of adults with MASH. In the current study, we propose critical roles for IFNγ signaling in macrophages and their release of IL-12 in modulating obesity-mediated insulin resistance in the liver and causing fatty liver progression to MASH phenotypes via hepatocyte IRS/FoxO1/FGF21 signaling (Fig. 9). While further studies are needed to test this hypothesis, our findings implicate that the IFNγ-IL12 axis may facilitate the intercellular crosstalk between macrophages and hepatocytes, playing a crucial role in the pathogenesis of metabolic liver disease. Our analysis of RNA-seq datasets from liver samples obtained from human subjects with MASLD and progressive steatohepatitis (MASH) and fibrosis[94] showed that expression of IFNγR1, IFNγR2, and IL-12B were significantly increased with advancing stages of liver fibrosis (F2 to F4) (Fig. 10A–D). The liver expression of FGF21 was also profoundly elevated in human subjects with F2-MASH fibrosis stage (Fig. 10E). These results signify the translational impact of our findings on human MASH. Moreover, a recent clinical study tested apilimod, a small-molecule inhibitor selectively suppressing IL-12 and IL-23, and found promising results in subjects afflicted with a common inflammatory disease, psoriasis[95]. Therefore, our findings pave the way for future development of small-molecule inhibitors selectively targeting IFNγ signaling and IL-12 as potential therapies to treat metabolic liver disease in humans.

## Methods

Our research complies with all relevant ethical regulations; Institutional Animal Care and Use Committee of the University of Massachusetts Chan Medical School.

### Animals, diet, and body composition analysis

IFNγR2-floxed mice (IFNγR2$^{f/f}$) were provided by Dr. Werner Muller, and LysM-Cre mice were provided by Dr. Roger Davis (University of Massachusetts Chan Medical School, Worcester, MA). LysM-Cre mice were crossed with IFNγR2-floxed mice to generate LysM-Cre$^+$-IFNγR2$^{f/f}$ (Lyz-IFNγR2$^{-/-}$) mice. Global IFNγ null mice and C57BL/6J mice were obtained from the Jackson Laboratory and studied at 4 - 5 months of age. All strains were on C57BL/6J background. Mice were housed under controlled temperature (23ºC) and lighting (12-hour light; 0700-1900 hour, 12-hour dark; 1900-0700 hour) with free access to water. Mice were fed a standard chow diet containing 14% kcal from fat, 60% kcal from carbohydrate, and 26% kcal from protein (LFD; LabDiet Prolab Isopro RMH3000 #5P75/5P76) or a high-fat diet (HFD) containing 60% kcal from fat, 26% kcal from carbohydrate, and 14% kcal from protein (Research Diets #D12492i, New Brunswick, NJ) ad libitum for a select duration as described in the text. For MASH studies, mice were fed a methionine-choline deficient (MCD) diet (A02082002BR, Research Diets, New Brunswick, NJ), a choline-deficient L-amino acid HFD (CDAHFD; HFD with 0.1% methionine; A06071302, Research Diets, New Brunswick, NJ), and Gubra Amylin NASH (GAN) diet (40% kcal from fat-palm oil, 20% kcal from fructose, and 2% cholesterol; D09100310, Research Diets, New Brunswick, NJ). Metabolic, analytical, and molecular experiments were performed in male Lyz-IFNγR2$^{-/-}$ mice and LysM-Cre$^+$ (WT) mice at 6 - 7 months of age following 10 weeks of an HFD or LFD as controls. Imaging and histological analyses and molecular experiments were performed in male Lyz-IFNγR2$^{-/-}$ mice and WT mice at 6 - 7 months of age after 4 weeks of an MCD diet, during 20 weeks of a CDAHFD starting at approximately 8 months of age, and during 20 weeks of a GAN diet starting at 6 months of age. For the IL-12 replacement study, metabolic, analytical, molecular, and imaging experiments were performed in male Lyz-IFNγR2$^{-/-}$ mice and WT mice fed an HFD for 12 weeks starting at 2 months of age. Whole-body fat and lean mass were noninvasively measured using $^1$H-Magnetic Resonance Spectroscopy (MRS) (Echo Medical Systems, Houston, TX). The current study examined male mice because female C57BL/6J mice are resistant to diet-induced obesity and insulin resistance, a primary endpoint of the current study. To study female mice, an ovariectomy procedure may be performed in the female C57BL/6J mice to become obese, but such a procedure creates other variables (e.g., lacking ovarian hormones) that may confound data interpretation. All animal studies complied with all relevant ethical regulations and were approved by the Institutional Animal Care and Use Committee of the University of Massachusetts Chan Medical School under protocol number #202000104.

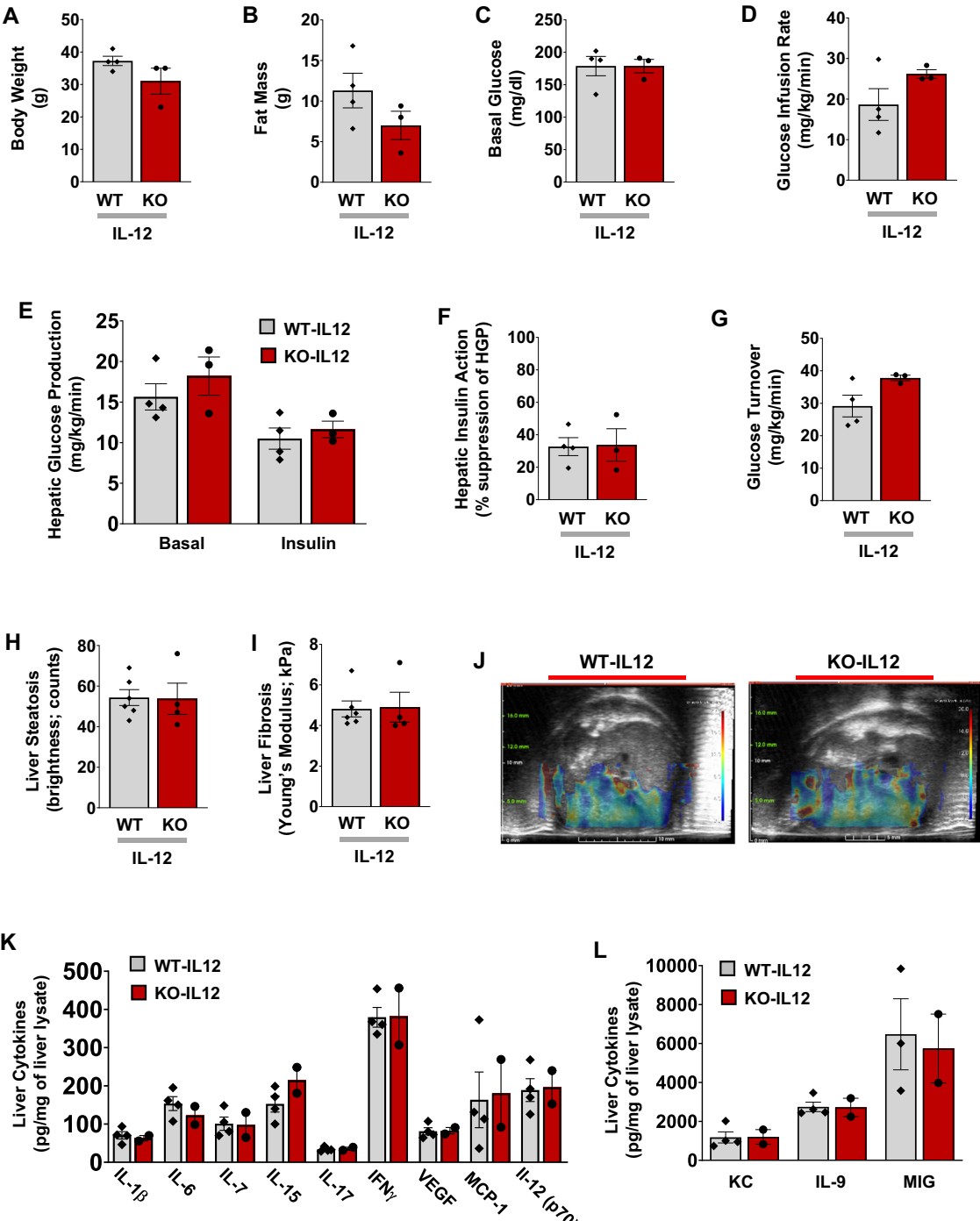

**Fig. 8 | IL-12 treatment re-establishes diet-induced insulin resistance, inflammation, and fibrosis in the liver of Lyz-IFNγR2⁻/⁻ mice.** Male Lyz-IFNγR2⁻/⁻ mice (KO; *n* = 3) and WT mice (*n* = 4) were fed an HFD for 12 weeks starting at 2 months of age, and a mouse recombinant IL-12 was chronically administered via osmotic pumps (1.2 ng/day/g body weight) during the last 2 weeks of HFD. A standardized hyperinsulinemic-euglycemic clamp was performed at the end to measure insulin action and glucose metabolism in awake mice. **A** Body weight in HFD-fed mice following IL-12 treatment. **B** Whole-body fat mass, measured using ¹H-MRS. **C** Basal plasma glucose levels after overnight fast. **D** Glucose infusion rate required to maintain euglycemia during the clamp. **E** Basal and clamp (insulin) hepatic glucose production (HGP). **F** Hepatic insulin action expressed as insulin-mediated percent suppression of basal HGP. **G** Whole-body glucose turnover. Additional cohorts of male Lyz-IFNγR2⁻/⁻ mice (KO; *n* = 4) and WT mice (*n* = 6) were fed a GAN diet for 8 weeks, and IL-12 was chronically administered via osmotic pumps (1.2 ng/day/g

body weight) during the last 2 weeks of the GAN diet. Vega Wide-field Ultrasound Imaging System was used to noninvasively assess liver steatosis and fibrosis, and multiplexed-Luminex was used to measure liver cytokines at the end. **H** Liver steatosis in GAN diet-fed mice after IL-12 treatment. **I** Liver fibrosis measured as stiffness using Young's Modulus. **J** Representative SWE images showing shear wave elasticity in regions of high and low intensity in mice. **K** Liver cytokines and chemokines, including IL-1β, IL-6, IL-7, IL-15, IL-17, IFNγ, VEGF, monocyte chemoattractant protein-1 (MCP-1), and IL-12 (p70) in GAN diet-fed mice after IL-12 treatment (*n* = 4 WT and 2 KO mice). **L** Liver cytokines (KC, IL-9, and MIG) in mice (*n* = 3 - 4 WT and 2 KO mice). Data are presented as mean ± SEM values. The statistical significance of the difference in mean values was determined using a two-tailed Student's *t*-test. For Fig. K & L, a two-tailed Mann-Whitney test was performed using GraphPad Prism (Version 10.2.0) for statistical analysis.

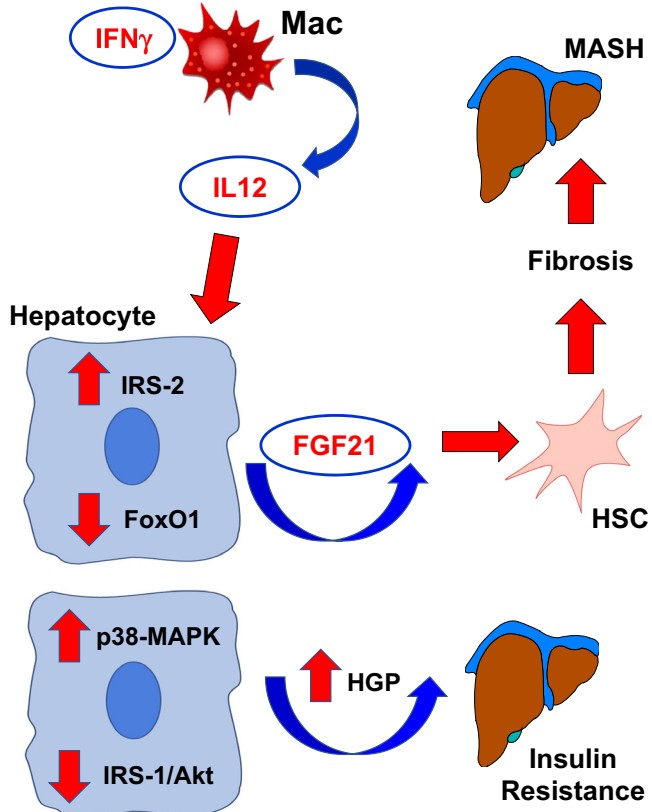

**Fig. 9 | Proposed model of the IFNγ-IL12 axis regulating liver insulin resistance and MASH.** Obesity activates macrophage IFNγ signaling, releasing a pro-inflammatory cytokine IL-12 that causes hepatic insulin resistance by down-regulating IRS/Akt-associated insulin signaling via p38-MAPK and induces fatty liver progression to MASH by modulating IRS/FoxO1/FGF21 pathway in the liver.

## Hyperinsulinemic-euglycemic clamp to assess insulin sensitivity and glucose metabolism

Survival surgery was performed in anesthetized mice to establish an indwelling catheter in the jugular vein. Mice were allowed to recuperate from surgery for 5 - 6 days before the clamp experiments. Following the overnight fast (-14 hours), a basal period of the experiment began with a continuous infusion of D-[3-³H]glucose (0.05 µCi/min) using microdialysis pumps to assess the basal rates of whole-body glucose turnover and hepatic glucose production (HGP)[44]. A blood sample (20 µl) was taken at the end of the basal period for the measurement of plasma glucose, insulin, and [³H]glucose concentrations (basal parameters). Following the basal period, a standard 2-hour hyperinsulinemic-euglycemic clamp was conducted with a primed and continuous infusion of human insulin (150 mU/kg body weight priming followed by 2.5 mU/kg/min; Novolin R; Novo Nordisk, Denmark) to raise plasma insulin within a physiological range (-0.8 ng/ml)[44]. Blood samples (10 µl) were collected at 10 - 20 min intervals for the immediate measurement of plasma glucose, and 20% dextrose (glucose) was infused at variable rates (glucose infusion rates) to maintain euglycemia ( - 120 mg/dl). During the clamp, [3-³H]glucose was continuously infused (0.1 µCi/min) to measure whole-body glucose turnover and glucose metabolic fluxes (glycolysis and glycogen plus lipid synthesis) and clamp HGP during the insulin-stimulated state[44]. The percent change from basal HGP to clamp HGP reflects hepatic insulin action as insulin-mediated percent suppression of HGP. To measure insulin-stimulated glucose uptake in individual organs, 2-[1-¹⁴C]deoxy-D-glucose (2-[¹⁴C]DG) was administered as a bolus (10 µCi) at 75 min after the start of the clamp[44]. Blood samples (20 µl) were taken at 80, 85, 90, 100, 110, and 120 min of the clamp for the measurement of

plasma [³H]glucose, ³H₂O, and 2-[¹⁴C]DG concentrations. Additional blood samples (20 µl) were taken at 120 min to measure plasma insulin concentrations (clamp parameters). At the end of the clamp, mice were anesthetized, and tissue samples, such as liver, skeletal muscle, and adipose tissue, were rapidly taken for biochemical and molecular analysis[44].

## IL-12 treatment in vivo

To examine the effects of IL-12 on glucose metabolism, a mouse recombinant IL-12 (0.25 µg/hour) or saline (control) was intravenously infused for 4 hours in male C57BL/6J mice. This was followed by a standardized hyperinsulinemic-euglycemic clamp with continuous IL-12 infusion in awake mice, as previously described[44]. For an acute IL-12 treatment, we collected liver samples 1 hour after IL-12 injection (1 mg) with saline serving as controls. For a chronic IL-12 treatment in HFD-fed mice (IL-12 replacement study), male Lyz-IFNγR2⁻/⁻ and WT mice were fed an HFD for 12 weeks, and a mouse recombinant IL-12 was chronically administered via osmotic pumps (1.2 ng/day/g body weight; Alzet, Cupertino, CA) during the last 2 weeks of HFD. For a chronic IL-12 treatment in GAN diet-fed mice, male Lyz-IFNγR2⁻/⁻ mice and WT mice were fed a GAN diet for 8 weeks, and IL-12 was chronically administered via osmotic pumps (1.2 ng/day/g body weight) during the last 2 weeks of the GAN diet.

## Biochemical analysis and calculation

Glucose concentrations during the clamp were measured with an Analox GM9 Analyzer (Analox Instruments Ltd., Hammersmith, London, UK) using 5 µl of plasma via glucose oxidase methodology. Plasma concentrations of [3-³H]glucose, 2-[¹⁴C]DG, and ³H₂O were determined following the deproteinization of plasma samples as previously described[44]. For the determination of tissue 2-[¹⁴C]DG-6-phosphate content, tissue samples were homogenized, and the supernatants were subjected to an ion-exchange column to separate 2-[¹⁴C]DG-6-P from 2-[¹⁴C]DG[44]. Rates of basal HGP and insulin-stimulated whole-body glucose turnover were determined as previously described[44]. HGP during an insulin-stimulated state was determined by subtracting the glucose infusion rate from whole-body glucose turnover. Whole-body glycolysis and glycogen plus lipid synthesis were calculated as previously described[44]. Insulin-stimulated glucose uptake in individual organs was measured by determining the tissue content of 2-[¹⁴C]DG-6-phosphate and plasma 2-[¹⁴C]DG profile[44].

## Liver triglyceride and hydroxyproline measurements

Liver samples (50 mg) were collected from mice and homogenized in 3:1 chloroform:methanol before incubation at room temperature with shaking for 4 hours. 1 M H₂SO₄ was added, and samples were shaken for 10 minutes before centrifugation. A total of 5 µl of the lower organic phase was processed for triglyceride (TG) content using a Sigma TG Kit (TR0100; Sigma, St. Louis, MI) following the manufacturer's instructions. For hydroxyproline measurements, liver samples were isolated from the same region of the left lobe, flash frozen, and used for homogenization. The liver homogenates were processed to quantify hydroxyproline concentrations using hydroxyproline assay kits and following the manufacturer's instructions (Sigma-Aldrich, MAK008).

## Molecular analysis of liver and isolated hepatocytes

For quantitative real-time PCR (RT-qPCR) to measure the liver expression of metabolic genes, RNA was isolated using TRIzol reagent (Invitrogen, Grand Island, NY) following the manufacturer's instructions. An Omniscript RT Kit (Qiagen, Valencia, CA) was used for cDNA synthesis, and samples were run on a CFX96 Real-Time System (BIO-RAD, Hercules, CA) using SYBR Green (BIO-RAD). The mRNA levels were normalized to *HPRT* as a housekeeping gene. Primer sequences are shown in Supp. Table 2. For liver protein analysis, tyrosine/serine/

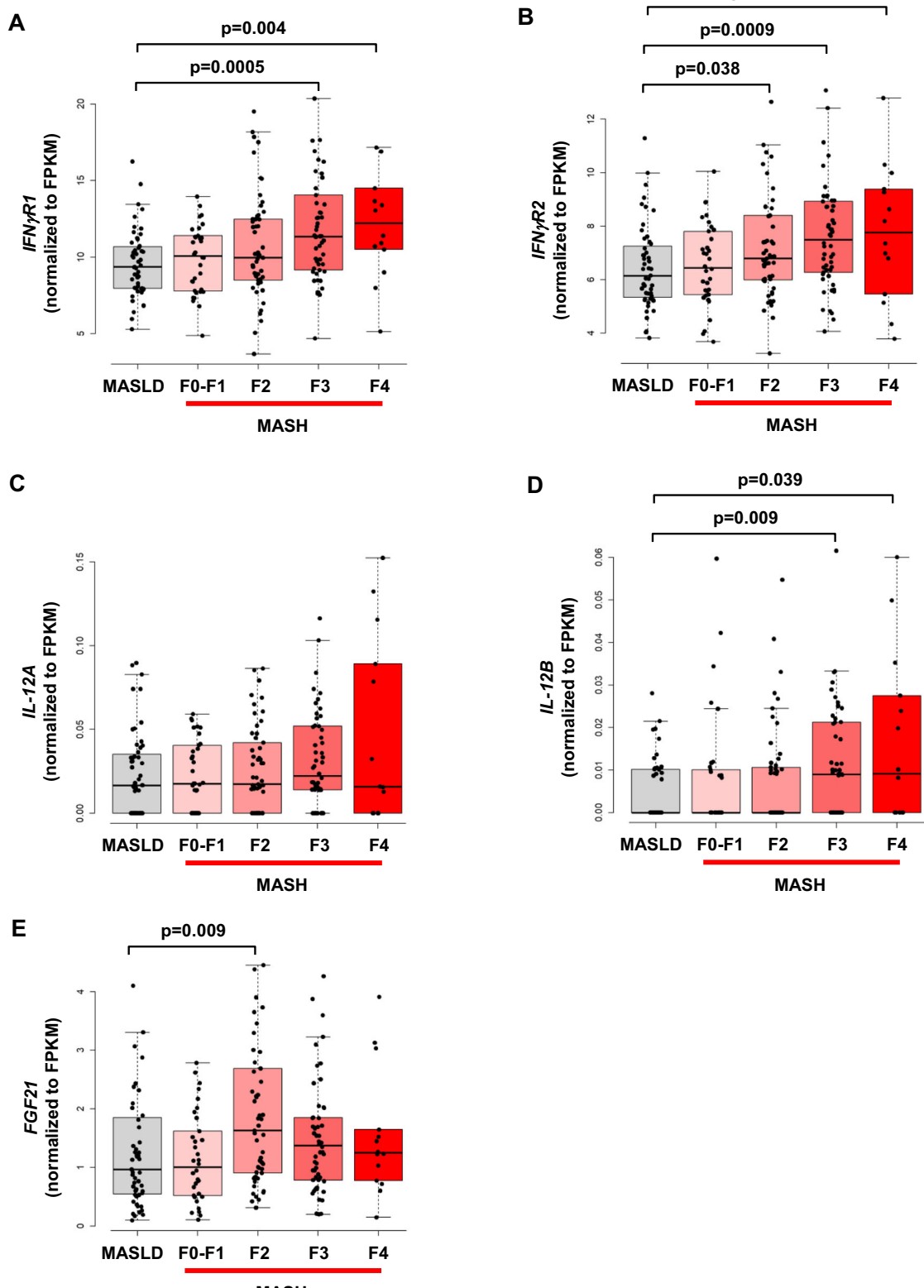

**Fig. 10 | IFNγ-IL12 axis is upregulated in human subjects with MASH.** PolyA-selected RNA-seq data from 206 patients diagnosed with MASLD/MASH from the NICBI GEO database (accession number: GSE135251) were analyzed to evaluate differential gene expression in human subjects with MASLD/MASH. The data were stratified into 5 groups: 51 patients with MASLD, 34 patients with MASH fibrosis stage F0-F1, 53 patients with F2-MASH, 54 patients with F3-MASH, and 14 patients with F4-MASH. **A** *IFNγR1*, **B** *IFNγR2*, **C** *IL-12A*, **D** *IL-12B*, and **E** *FGF21*. The counts were normalized to Fragments Per Kilobase of transcript per Million (FPKM). The statistical significance of the difference in mean values was determined using a two-sided Wilcoxon-Mann-Whitney test. Lower bound (Q1): 25%, upper bound (Q3): 75%, and centre (median): 50%. Interquartile Range (IQR): Q3-Q1. Minima (lower whisker end): Q1-1.5xIQR. Maxima (upper whisker end): Q3 + 1.5xIQR.

threonine phosphorylated protein and total protein levels were measured using the Jess Multiplexing Western Blot System (ProteinSimple, Bio-techne, Minneapolis, MN). The protein levels were normalized to β-actin as a loading control. Following a 4-hour intravenous infusion of IL-12 (0.25 µg/hour) or saline (control) in WT mice, primary hepatocytes were isolated via Percoll gradient, yielding ~1.5×10⁶ cells per mouse, and the wells were seeded with 0.5×10⁶ cells in triplicates for RT-qPCR analysis[96].

### Multiplexed-Luminex analysis and ELISA
Cytokine and chemokine levels in serum and detergent-free liver tissue lysates were analyzed with a Bio-Plex 200 Luminex system (BIO-RAD) using MILLIPLEX® MAP kits (Millipore, Bedford, MA) according to manufacturer's instructions. Plasma FGF21 levels were measured using an ELISA kit (R&D Systems, Minneapolis, MN).

### Noninvasive analysis of liver steatosis and fibrosis using 3D-ultrasound imaging
We used a state-of-the-art Vega Wide-field Ultrasound Imaging System (SonoVol, Inc., Durham, NC) to noninvasively assess liver steatosis and fibrosis in mice during MASH diets. Mice were anesthetized using isoflurane and laid on a pre-warmed membrane with a continuous flow of gas anesthesia. Once the mouse was properly positioned, an initial 3D B-Mode (conventional ultrasound) image was taken to obtain an image of the tissue borders for anatomical context. Using multiple views (transverse, lateral, sagittal), a target location within the liver was identified, and the transducer was directed to that position for shear wave elastography (SWE) imaging, which was used to noninvasively measure tissue stiffness. By pushing on the tissue with high-frequency ultrasound waves at a specific location, we measured the stiffness (m/s, kPa) and brightness (counts) of the liver. Stiffness represented liver fibrosis, and brightness represented liver steatosis. Multiple SWE images ( ~ 4 total) were obtained throughout the liver (left and right lobes) of each mouse to obtain a broad sample set. The color-mapped SWE overlay on grayscale B-Mode provided a window into the organ's mechanical properties at that location. During analysis, each image was carefully assessed, and only homogenous liver tissues were measured. Artifacts were removed, and threshold/confidence floors were applied.

### Histological analysis and pathology evaluation of the liver
Liver samples were collected from Lyz-IFNγR2⁻/⁻ and WT mice at the end of respective MASH diets for histological analysis. Freshly extracted liver samples were fixed in 10% formalin for 72 hours and embedded in paraffin blocks. Sections (5 µm) were cut for hematoxylin-eosin (H&E) staining and Masson's Trichrome staining with collagen stained in blue. Sections were also stained for α−SMA. Sample slides were provided to Applied Pathology Systems (Shrewsbury, MA) for a complete pathology and MAS evaluation (e.g., grading of steatosis, ballooning and lobular inflammation, hepatocellular injury, fibrosis) by a board-certified pathologist. The total pathology scores were a sum of the portal (P), lobular (L), and perivenular (pV) scores for inflammatory infiltrate, macro-vesicular (MacV) and micro-vesicular (MicV) scores for steatosis, ballooning (Bal), acidophilic body (AB), mallory body, and necrosis scores for hepatocellular injury, bile duct injury scores, and portal, pericellular (PC), bridging (Br), and cirrhosis (C) scores for fibrosis (Supp. Table 1).

### RNA-seq data analysis of liver genes in human subjects with MASLD and MASH
To investigate gene expression in human subjects with metabolic liver disease, we analyzed polyA-selected RNA-seq datasets from liver samples obtained from 206 patients diagnosed with MASLD and progressive steatohepatitis (MASH) and fibrosis from the NICBI GEO database (accession number: GSE135251)[94,97]. These patients were stratified into 5 groups according to MASLD and MASH fibrosis stages

described in the dataset: 51 patients with MASLD, 34 patients with MASH fibrosis stages F0-F1, 53 patients with MASH fibrosis stage F2, 54 patients with MASH fibrosis stage F3, and 14 patients with MASH fibrosis stage F4. For gene quantification, we employed the GENCODE (v42) GTF file and FeatureCounts to calculate locus-specific read counts[98]. These counts were normalized to Fragments Per Kilobase of transcript per Million (FPKM), accounting for both gene exon lengths and sample-specific unique-mapped read counts. Mapping to all annotated isoforms in GENCODE (v42) of the indicated genes (IFNgR1, IFNgR2, IL-12A, IL-12B, and FGF21) was analyzed for MASLD and each MASH group[99]. Statistical significance was assessed using a two-sided Wilcoxon-Mann-Whitney test.

### Statistical analyses
All data are expressed as mean ± SEM values. The statistical significance of the difference in mean values was determined using a one-way analysis of variance (ANOVA) with Tukey's multiple comparison test for post-hoc analysis and a two-tailed Student's t-test. A probability value of $p < 0.05$ was used as the criterion for statistical significance. All analyses were performed using Statistical Analysis Software (SAS, Inc., Cary, NC).

### Reporting summary
Further information on research design is available in the Nature Portfolio Reporting Summary linked to this article.

## Data availability
The minimum datasets generated and/or analyzed during the current study that are necessary to interpret, verify, and extend the research in the article are available from the corresponding author. All other data needed to reproduce the results presented here can be found in the manuscript, figures, and supplementary material. Previously published datasets of human liver tissue were obtained from GSE135251, accessible on the NCBI GEO repository. All scientific data and metadata generated in this study will be deposited in the National Mouse Metabolic Phenotyping Center database that will be accessible to the public upon creation of a user account. Source data are provided with this paper.

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

## Acknowledgements

We would like to thank Dr. Roger Davis (H. Arthur Smith Professor and Chair of Molecular Medicine at the University of Massachusetts Chan Medical School) for the kind donation of lysozyme 2 (LysM)-Cre mice and the discussion in creating Lyz-IFNγR2$^{-/-}$ mice. We appreciate the technical assistance from Dr. Mohammed Salman Shazeeb, Associate Professor and Director of Image Processing & Analysis Core (iPAC) at the University of Massachusetts Chan Medical School, as a portion of the image analysis was performed using resources from the iPAC. We also appreciate the intellectual input from Dr. Ozkan Aydemir (Assistant Professor and Director of Bioinformatics at the University of Massachusetts Chan Medical School) for biostatistics and bioinformatics analysis. This study was supported by grants from the National Institutes of Health (R01DK133772 awarded to JKK; R01DK116999 awarded to ACM; R03ED032455 and UL1TR001453 awarded to CZ).

## Author contributions

R.H.F. and J.K.K. conceived of, designed and wrote the manuscript for this project. J.K.K., C.Z., A.C.M., and K.W.L. provided supervision and project administration and secured financial support. R.H.F., H.L.N., S.Su., M.A., S.D., B.K., B.-Y.K., S.D.R., and C.Z. led animal, tissue and cell-based, in vivo, metabolic, biochemical, and molecular experiments and data analyzes. R.H.F., H.L.N., S.Su., M.A., S.D., S.Sa., B.K., A.M.K., L.H.K., L.A.T., N.M.B.T., S.C., B.-Y.K., S.D.R., K.K., C.S., B.J.T., C.Z., Z.L., V.M.B., P.R.P., D.X.T.Z., K.I., A.B., X.H., and D.A.T. performed animal, tissue and cell-based, in vivo, metabolic, biochemical, and/or molecular experiments and data analyzes. R.H.F., H.L.N., S.Su., M.A., S.D., S.Sa., B.K., A.M.K., L.H.K., L.A.T., N.M.B.T., S.C., B.-Y.K., S.D.R., K.K., C.S., B.J.T., C.Z., Z.L., V.M.B., P.R.P., D.X.T.Z., K.I., A.B., X.H., D.A.T., W.M., D.L.G., A.C.M., K.W.L., and J.K.K. reviewed the manuscript for intellectual content. R.H.F., H.L.N., S.Su., M.A., S.D., B.K., L.A.T., B.-Y.K., S.D.R., C.Z., W.M., D.L.G., A.C.M., K.W.L., and J.K.K. revised the manuscript for intellectual content.

## Competing interests

J.K.K. is on the scientific advisory board for Elevian Inc. and Imagine Pharma. D.L.G. is a consultant for The Jackson Laboratory. The remaining authors declare no competing interests.
