## [Peer Review File · Nature Communications]

IFN γ -IL12 axis regulates intercellular crosstalk in metabolic dysfunction-associated steatotic liver diseaseREVIEWER COMMENTS

Reviewer #1 (Remarks to the Author):

In this study, Friedline and coworkers examined the impact of IFN γ R2 specific Ko in myeloid cells (Lyz-IFN γ R2 $^{-/-}$) on nonalcoholic fatty liver disease (NAFLD) features in experimental models in mice. They found that Lyz-IFN γ R2 $^{-/-}$ mice are protected against insulin resistance upon the development of NAFLD under high fat diet (HFD), which was associated with reduced intrahepatic synthesis of IL-12. They show that IL-12 administration can cause hepatic insulin resistance. Upon feeding a nonalcoholic steatohepatitis (NASH) inducing diet, Lyz-IFN γ R2 $^{-/-}$ mice are protected against fibrosis, a phenotype associated with increased Foxo1 and reduced Fgf21 levels.

Major comments:

- Results presented in Figure 1 are not fully consistent with the subsequent data obtained in Lyz-IFN γ R2 $^{-/-}$ mice and their wild-type controls: if IFN γ is mediating hepatic insulin resistance through IFN γ R2 the full IFN γ ko should also have improved hepatic insulin resistance. It is noted that the duration of HFD was different between the two models (6 vs. 10 weeks) rendering the results more difficult to compare. The animal strain for the IFN γ ko is not reported in the methods; is it C57/Bl6 as for Lyz-IFN γ R2 $^{-/-}$ mice? What do Figure 1 data add to the manuscript message? They seem to me they raise more questions than they those they answer.
- Hepatic activation status of InsR/Akt/FoxO1 signaling (a main driver of hepatic G6Pc/Pepck transcriptional regulation in response to insulin, see Fig. 2k) in Fig. 2 model? This is especially important as FoxO1 activation is reported later in the paper.
- The same should be reported after IL-12 administration (Fig. 3); this is partly done for P-Tyr-IRS1, but not statistically significant \diamond no conclusion can be drawn.
- To support a role of IL-12 in mediating protection against insulin resistance in Lyz-IFN γ R2 $^{-/-}$ mice, authors should demonstrate whether IL-12 infusion can revert insulin sensitivity in this model as well.
- Claims about a protection of Lyz-IFN γ R2 $^{-/-}$ mice against NASH development should be supported by the detailed histological analysis with grading of steatosis, ballooning and lobular inflammation, plus histological staining of liver fibrosis (ideally quantification of

fibrosis with Oh-proline measurement, activation of hepatic stellate cells with asma/tgf-beta stainings). I do acknowledge reduction of animals needed for each experiment is an important goal of experimental research, but when showing noninvasive approaches these need to be validated against the gold standard in the literature. In addition, individual data and high quality histological samples (or at least a table with complete histological scorings for each animal) should be shown (current images are just “representative” impossible to be observed in detail. What is the impact of acute IL-12 infusion in re-establishing hepatic inflammation in this model?

- Figure 5: Authors should show Foxo1 protein levels and Foxo1 phosphorylation status, mRNA levels alone are not informative. Plus Foxo1 is supposed to induce hepatic insulin resistance (there is ample literature on that) so how does IL-12 downregulation of Foxo1 fit with the previous observations? What’s the link with liver disease progression?
- Please report in Figure of legends the number of animals used in each experiment.
- The study findings would be strongly reinforced by validation of upregulation of IFN γ R2 and IL-12 during NAFLD in human liver datasets (several can be tested as publicly available or upon collaboration with clinical researchers), and correlation of their expression levels with insulin resistance and fibrosis.

Minor:

- Please amend the introduction: there is no (not yet at least) human genetics evidence that IFN is implicated in the pathogenesis of NAFLD; refs 8,17 are just to general reviews of the field, please amend or report robust genetic association data, or report original experimental studies specifying evidence is limited to rodents.

Reviewer #2 (Remarks to the Author):

In the study by Friedline et al., the authors investigated the role of the IFN γ -IL12 axis in regulating intercellular communication in the liver and its potential as a therapeutic target for treating non-alcoholic fatty liver disease (NAFLD). The primary data for this study were obtained from Lyz-IFN γ R2-/- mice and wild type (WT) mice fed with diets to induce NASH. Although the study provides some valuable insights, some of the data presented are not sufficient to draw definitive conclusions, and further experimental validation is required.

1. To gain a more comprehensive understanding of the role of IFN γ signaling in myeloid cells in NASH, particularly the interaction between macrophages and hepatocytes, single-cell RNA-seq should be performed on liver samples from Lyz-IFN γ R2 $^{-/-}$ and WT mice with NASH. Alternatively, RNA-seq of primary hepatocytes and macrophages isolated from these mice separately could be performed.
2. While the authors showed that IL12 regulates hepatocyte expression of IRS-2 and FoxO1, and that FGF21 secreted by hepatocytes promotes fibrosis via HSC in the liver, it is unclear if IL12 plays a direct role in promoting HSC in fibrosis. In addition, protein levels should be examined to validate this finding, as the authors mainly checked relative gene expression in mRNA levels in Figure 5. In vitro experiment using macrophages isolated from liver could be performed. The IL12 levels in the macrophage supernatant should be measured, followed by coculture the supernatant with primary hepatocytes to validate the findings in Figure 5.
3. Successful IFN γ knockout in myeloid cells should be confirmed. Additionally, it is essential to use myeloid cells IFN γ knockin mice in NASH models.
4. As the authors identified the IFN γ -IL12 axis as a potential therapeutic target for treating NAFLD, it is important to investigate if any drugs or small molecule inhibitors are available for clinical translation.
5. The inclusion of histological scores for NASH in each NASH model would be helpful.
6. A limitation of this study is the lack of data from NASH patients to support the findings.
7. Grammar could be improved.

Point-by-Point Response to Reviewers' Comments for NCOMMS-23-22157-T

Reviewer #1:

Major comments:

1. *Results presented in Figure 1 are not fully consistent with the subsequent data obtained in Lyz-IFN γ R2^{-/-} mice and their wild-type controls: if IFN γ is mediating hepatic insulin resistance through IFN γ R2 the full IFN γ ko should also have improved hepatic insulin resistance. It is noted that the duration of HFD was different between the two models (6 vs. 10 weeks) rendering the results more difficult to compare. The animal strain for the IFN γ ko is not reported in the methods; is it C57/Bl6 as for Lyz-IFN γ R2^{-/-} mice? What do Figure 1 data add to the manuscript message? They seem to me they raise more questions than they those they answer.*

The global IFN γ ^{-/-} mice obtained from The Jackson Laboratory were on C57BL/6J background, and those data supported our notion that IFN γ as a cytokine has pleiotropic effects in other organs while our manuscript focuses on the macrophage IFN γ signaling with downstream cytokines. Nonetheless, we agree with the reviewer that these data add minimally to the manuscript and therefore, have removed the relevant figure in the revision.

2. *Hepatic activation status of InsR/Akt/FoxO1 signaling (a main driver of hepatic G6Pc/Pepck transcriptional regulation in response to insulin, see Fig. 2k) in Fig. 2 model? This is especially important as FoxO1 activation is reported later in the paper.*

As suggested by the reviewer, we used the Jess Multiplexed Western Blot System to perform a new molecular analysis of liver insulin signaling in Lyz-IFN γ R2^{-/-} and WT mice after HFD. We found that liver Ser⁴⁷³-phosphorylation of Akt was significantly increased by 20% in Lyz-IFN γ R2^{-/-} mice compared to WT mice, indicating improved hepatic Akt activity. Liver FoxO1 protein levels were dramatically decreased by 97% in Lyz-IFN γ R2^{-/-} mice, and this is consistent with insulin stimulation of Akt that phosphorylates FoxO1, causing its translocation out of the nucleus and degradation, which results in suppression of gluconeogenic genes. These results are also consistent with reduced HGP and improved hepatic insulin action in HFD-fed Lyz-IFN γ R2^{-/-} mice. Liver InsR protein levels were not altered in Lyz-IFN γ R2^{-/-} mice, suggesting altered regulation of insulin signaling downstream of InsR. The new data are shown in Fig. 1K and discussed on p.6.

3. *The same should be reported after IL-12 administration (Fig. 3); this is partly done for P-Tyr-IRS1, but not statistically significant \diamond no conclusion can be drawn.*

We also performed a new molecular analysis of liver insulin signaling in WT mice after IL-12 treatment in vivo and found 36~57% decreases in liver IRS-1, Akt, and Ser⁴⁷³-phosphorylation of Akt protein levels in IL12-treated mice compared to controls. These results are consistent with IL-12 effects to increase HGP and impair hepatic insulin action. The new data are shown in Fig. 3 and discussed on p.7-8.

4. *To support a role of IL-12 in mediating protection against insulin resistance in Lyz-IFN γ R2^{-/-} mice, authors should demonstrate whether IL-12 infusion can revert insulin sensitivity in this model as well.*

We appreciate the reviewer's suggestion and have performed new experiments upon breeding additional cohorts of Lyz-IFN γ R2^{-/-} and WT mice. Male Lyz-IFN γ R2^{-/-} (KO) and WT mice were fed an HFD for 12 weeks, and a mouse recombinant IL-12 was chronically administered via osmotic pumps (1.2 ng/day/g body weight; Alzet) during the last 2 weeks of HFD. A chronic IL-12 treatment re-established hepatic insulin resistance and type 2 diabetes phenotypes in HFD-fed Lyz-IFN γ R2^{-/-} mice that were previously shown to be protected from diet-induced insulin resistance in the liver. During the hyperinsulinemic-euglycemic clamp, HGP was minimally suppressed by insulin in Lyz-IFN γ R2^{-/-} mice and to a level comparable to the WT mice. As a result, hepatic insulin action was measured at ~30% for both groups of HFD-fed mice following IL-12 treatment, and this was profoundly lower than the 77% hepatic insulin action previously measured in HFD-fed Lyz-IFN γ R2^{-/-} mice. Thus, these results indicate that Lyz-IFN γ R2^{-/-} mice lost their protection from diet-induced insulin resistance in the liver following a chronic IL-12 treatment. The new data are shown in Fig. 7 and discussed on p.11.

5. *Claims about a protection of Lyz-IFN γ R2^{-/-} mice against NASH development should be supported by the detailed histological analysis with grading of steatosis, ballooning and lobular inflammation, plus histological staining of liver fibrosis (ideally quantification of fibrosis with Oh-proline measurement, activation of hepatic stellate cells with α 1/2-tgf-beta stainings). I do acknowledge reduction of animals needed for each experiment is an important goal of experimental research, but when showing noninvasive approaches these need to be validated against the gold standard in the literature. In addition, individual data and high quality histological samples (or at least a table with complete histological scorings for each animal) should be shown (current images are just "representative" impossible to be observed in detail. What is the impact of acute IL-12 infusion in re-establishing hepatic inflammation in this model?)*

We thank the reviewer for constructive suggestions. We performed a detailed histological analysis of liver samples collected from Lyz-IFN γ R2^{-/-} and WT mice following the GAN diet, a physiological obesogenic diet that promotes liver lesions with morphological characteristics closely resembling human NASH (ref. 60-62). This analysis involved a comprehensive pathology evaluation of liver sections with H&E and Masson's Trichrome stain by a board-certified pathologist (Applied Pathology Systems) for the grading of inflammatory infiltrate, steatosis, hepatocellular injury, bile duct injury, and fibrosis. We found that there was an across-the-board decrease in the histology scores for inflammatory infiltrate, hepatocellular injury, and fibrosis in Lyz-IFN γ R2^{-/-} mice compared to WT mice following the GAN diet whereas the histology scores for steatosis were comparable between both groups of mice. As a result, the total pathology scores were significantly reduced in Lyz-IFN γ R2^{-/-} mice compared to WT mice after 20 weeks of the GAN diet, and this is consistent with liver steatosis and fibrosis data obtained from the ultrasound imaging. The complete histology scorings for each animal are shown in Table 1, and representative histology images are shown in Fig. 5. Additionally, we processed the liver sections for α -SMA and TGF- β stain and found a 71% decrease in

percent coverage of α -SMA stain in GAN diet-fed Lyz-IFN γ R2^{-/-} mice compared to WT mice, indicating attenuated HSC activation in Lyz-IFN γ R2^{-/-} mice. The TGF- β stain did not show significant effects in Lyz-IFN γ R2^{-/-} mice (see below images). The new data with representative images from α -SMA stain for 2 WT and 2 KO mice are shown in Fig. 5. Lastly, we used a colorimetric assay to measure total hydroxyproline levels in the liver and found no differences between Lyz-IFN γ R2^{-/-} and WT mice following the GAN diet (see below figure). In that regard, Gomes et al. have recently reported that high variability in hydroxyproline levels within a single liver with fibrosis may limit its value in quantifying the extent of fibrosis (ref). All of these new results are discussed on p.9.

Gomes, A. T., Bastos, C. G., Afonso, C. L., Medrado, B. F. & Andrade, Z. A. How variable are hydroxyproline determinations made in different samples of the same liver? *Clin Biochem* **39**, 1160-1163, doi:10.1016/j.clinbiochem.2006.08.002 (2006).

Moreover, we conducted new animal experiments in which male Lyz-IFN γ R2^{-/-} mice (KO) and WT mice were fed a GAN diet for 8 weeks, and IL-12 was chronically administered via osmotic pumps (1.2 ng/day/g body weight) during the last 2 weeks of the GAN diet. Using Vega Wide-field Ultrasound Imaging System, we found that GAN diet-fed Lyz-IFN γ R2^{-/-} and WT mice developed a comparable degree of liver steatosis and fibrosis after IL-12 treatment, suggesting that IL-12 re-established fatty liver progression to liver fibrosis in Lyz-IFN γ R2^{-/-} mice. We also performed multiplexed-analysis of cytokines using Luminex in liver samples collected at the end of the GAN diet and IL-12 treatment. Many of the pro-inflammatory cytokines/chemokines were similarly elevated in the livers of GAN diet-fed Lyz-IFN γ R2^{-/-} and WT mice after IL-12 treatment, indicating that IL-12 also re-established hepatic inflammation in Lyz-IFN γ R2^{-/-} mice. The new data are shown in Fig. 7 and discussed on p. 11.

6. *Figure 5: Authors should show Foxo1 protein levels and Foxo1 phosphorylation status, mRNA levels alone are not informative. Plus Foxo1 is supposed to induce hepatic insulin resistance (there is ample literature on that) so how does IL-12 downregulation of Foxo1 fit with the previous observations? What's the link with liver disease progression?*

Using the Jess Multiplexed Western Blot System, we performed a new molecular analysis to measure liver FoxO1 protein levels in different mouse models. Our data indicate that liver FoxO1 protein levels were dramatically decreased by 97% in Lyz-IFN γ R2^{-/-} mice compared to WT mice after HFD, and these results are consistent with decreases in gluconeogenic genes and clamp HGP in HFD-fed Lyz-IFN γ R2^{-/-} mice. We also found that

IL-12 treatment caused a 41% decrease in liver FoxO1 protein levels, consistent with a 20% decrease in *FoxO1* mRNA levels in the liver of IL12-treated mice compared to controls. This may be due to IL-12 activation of p38-MAPK, which was shown to regulate the transcriptional activity of FoxO1 via phosphorylation (ref. 69). Deng et al. have also found that pro-inflammatory cytokines, IL-2, IL-12, and IL-15, induce phosphorylation of FoxO1, leading to its inactivation in NK cells, and IL-12 was shown to promote the highest level of FoxO1 phosphorylation after 2 hours of treatment (ref. 70). Thus, hepatocyte transcription factor FoxO1 may be competitively regulated by insulin and pro-inflammatory cytokines, possibly leading to different pathogenic outcomes for the liver. The new data are shown in Fig. 1K and 6C and discussed on p.6 and 10.

7. Please report in Figure of legends the number of animals used in each experiment.

We apologize for this oversight, and the number of animals used in each experiment has now been added to all figure legends.

8. The study findings would be strongly reinforced by validation of upregulation of IFN γ R2 and IL-12 during NAFLD in human liver datasets (several can be tested as publicly available or upon collaboration with clinical researchers), and correlation of their expression levels with insulin resistance and fibrosis.

We appreciate the reviewer's suggestion. However, we feel that such data from human NASH liver does not establish a causal relationship between the IFN γ -IL12 axis and metabolic liver disease, which was the focus of the current study. We have added a recent cross-sectional study that has found a strong association between serum IL-12 levels and the severity of NAFLD progression (ref. 30), and this is now discussed on p.3.

Minor:

1. Please amend the introduction: there is no (not yet at least) human genetics evidence that IFN is implicated in the pathogenesis of NAFLD; refs 8,17 are just to general reviews of the field, please amend or report robust genetic association data, or report original experimental studies specifying evidence is limited to rodents.

We have deleted this sentence in the revision.

Reviewer #2:

1. *To gain a more comprehensive understanding of the role of IFN γ signaling in myeloid cells in NASH, particularly the interaction between macrophages and hepatocytes, single-cell RNA-seq should be performed on liver samples from Lyz-IFN γ R2 $^{-/-}$ and WT mice with NASH. Alternatively, RNA-seq of primary hepatocytes and macrophages isolated from these mice separately could be performed.*

We appreciate the reviewer's suggestion. However, due to our limited budget and the prohibitive cost of the RNA-seq analysis, we believe that this is beyond the scope of the current study.

2. *While the authors showed that IL12 regulates hepatocyte expression of IRS-2 and FoxO1, and that FGF21 secreted by hepatocytes promotes fibrosis via HSC in the liver, it is unclear if IL12 plays a direct role in promoting HSC in fibrosis. In addition, protein levels should be examined to validate this finding, as the authors mainly checked relative gene expression in mRNA levels in Figure 5. In vitro experiment using macrophages isolated from liver could be performed. The IL12 levels in the macrophage supernatant should be measured, followed by coculture the supernatant with primary hepatocytes to validate the findings in Figure 5.*

As suggested by the reviewer, we used the Jess Multiplexed Western Blot System to perform a new molecular analysis to measure liver FoxO1 protein levels in different mouse models. Our data indicate that IL-12 treatment in vivo caused a 41% decrease in liver FoxO1 protein levels, consistent with a 20% decrease in FoxO1 mRNA levels in the liver of IL12-treated mice compared to controls. This may be due to IL-12 activation of p38-MAPK, which was recently shown to regulate the transcriptional activity of FoxO1 via phosphorylation (ref. 69). Deng et al. have also found that pro-inflammatory cytokines, IL-2, IL-12, and IL-15, induce phosphorylation of FoxO1, leading to its inactivation in NK cells, and IL-12 was shown to promote the highest level of FoxO1 phosphorylation after 2 hours of treatment (ref. 70). Thus, hepatocyte transcription factor FoxO1 may be competitively regulated by insulin and pro-inflammatory cytokines, possibly leading to different pathogenic outcomes for the liver. The new data are shown in Fig. 6C and discussed on p.10.

Moreover, the causal role of IL-12 as a macrophage-derived cytokine downstream of IFN γ signaling to modulate fatty liver progression to NASH was examined using a new cohort of Lyz-IFN γ R2 $^{-/-}$ mice that were previously shown to be protected from developing liver fibrosis after NASH diets. Male Lyz-IFN γ R2 $^{-/-}$ mice (KO) and WT mice were fed a GAN diet for 8 weeks, and IL-12 was chronically administered via osmotic pumps (1.2 ng/day/g body weight; Alzet) during the last 2 weeks of the GAN diet. Using Vega Wide-field Ultrasound Imaging System, we found that Lyz-IFN γ R2 $^{-/-}$ and WT mice developed a comparable degree of liver steatosis and fibrosis following the GAN diet and IL-12 treatment, indicating that IL-12 re-established fatty liver progression to liver fibrosis in Lyz-IFN γ R2 $^{-/-}$ mice. We also performed multiplexed-analysis of cytokines using Luminex in liver samples collected at the end of the GAN diet and IL-12 treatment. Many of the pro-inflammatory cytokines/chemokines were similarly elevated in the livers of Lyz-IFN γ R2 $^{-/-}$ and WT mice after the GAN diet and IL-12 treatment. Thus, a chronic IL-12 treatment re-

established the effects of the GAN diet to induce liver inflammation and fibrosis in Lyz-IFN γ R2^{-/-} mice, implicating a pivotal role for IL-12 in metabolic liver disease. The new data are shown in Fig. 7 and discussed on p.11.

3. Successful IFN γ knockout in myeloid cells should be confirmed. Additionally, it is essential to use myeloid cells IFN γ knockin mice in NASH models.

We appreciate the reviewer's insightful suggestions. However, the current study focused on IFN γ signaling in myeloid cells using conditional deletion of IFN γ R2 in our Lyz-IFN γ R2^{-/-} mice. While the myeloid cells IFN γ knockin mice would provide additional insights, given the time frame and our limited resources, we aren't able to generate such mice for this revision. We hope that the new data generated from an extensive list of new experiments further support our conclusion.

4. As the authors identified the IFN γ -IL12 axis as a potential therapeutic target for treating NAFLD, it is important to investigate if any drugs or small molecule inhibitors are available for clinical translation.

Our search has revealed that while there are antagonists and small-molecule inhibitors for type 1 interferons, we could not find any drugs or small-molecule inhibitors for type 2 interferon γ . We also could not find any drugs or small-molecule inhibitors selective for IL-12, but we were able to find apilimod, a small-molecule inhibitor suppressing IL-12 and IL-23 synthesis, which was recently shown to improve psoriasis condition in humans (ref. 87). Thus, our findings pave the future development of small-molecule inhibitors selectively targeting IFN γ signaling and IL-12 as potential therapies to treat NASH in humans. This is now discussed on p.12.

5. The inclusion of histological scores for NASH in each NASH model would be helpful.

We thank the reviewer for constructive suggestions. We performed a detailed histological analysis of liver samples collected from Lyz-IFN γ R2^{-/-} and WT mice following the GAN diet, a physiological obesogenic diet that promotes liver lesions with morphological characteristics closely resembling human NASH (ref. 60-62). This analysis involved a comprehensive pathology evaluation of liver sections with H&E and Masson's Trichrome stain by a board-certified pathologist (Applied Pathology Systems) for the grading of inflammatory infiltrate, steatosis, hepatocellular injury, bile duct injury, and fibrosis. We found that there was an across-the-board decrease in the histology scores for inflammatory infiltrate, hepatocellular injury, and fibrosis in Lyz-IFN γ R2^{-/-} mice compared to WT mice following the GAN diet whereas the histology scores for steatosis were comparable between both groups of mice. As a result, the total pathology scores were significantly reduced in Lyz-IFN γ R2^{-/-} mice compared to WT mice after 20 weeks of the GAN diet, and this is consistent with liver steatosis and fibrosis data obtained from the ultrasound imaging. The complete histology scorings for each animal are shown in Table 1, and representative histology images are shown in Fig. 5. Moreover, we processed the liver sections for α -SMA and found a 71% decrease in percent coverage of α -SMA stain in GAN diet-fed Lyz-IFN γ R2^{-/-} mice compared to WT mice, indicating attenuated HSC activation in Lyz-IFN γ R2^{-/-} mice. The new data with representative images from α -SMA

stain for 2 WT and 2 KO mice are shown in Fig. 5. All of these new results are discussed on p.9.

6. *A limitation of this study is the lack of data from NASH patients to support the findings.*

We appreciate the reviewer's feedback. However, we feel that such data from human NASH liver does not establish a causal relationship between the IFN γ -IL12 axis and metabolic liver disease, which was the focus of the current study. We have added a recent cross-sectional study that has found a strong association between serum IL-12 levels and the severity of NAFLD progression (ref. 30), and this is now discussed on p.3.

7. *Grammar could be improved.*

We apologize for this oversight, and we have thoroughly edited for grammar using Grammarly software and other proofreaders for this revision.

REVIEWER COMMENTS

Reviewer #1 (Remarks to the Author):

Authors should be commended as they have performed a new series of experiments to address my comments, in particular chronic IL12 infusion and systematic assessment of liver histology, that overall support the main study findings and strengthen the conclusions.

However, I think there are some remaining issues that need to be addressed:

- response to point 1: here Authors have accumulated several different concerns, but answered only to the last one! Please address the others as well (at least limitations and findings that still do not fit with the Authors' hypothesis must be discussed)
- hydroxyproline content measurement is still considered the gold standard for fibrosis assessment; it's true there may be some sampling variability, but this is also applies and is even more marked for liver histology, especially given the differences in staining intensity in the "representative samples" shown. Fibrosis cannot be assessed by ultrasonography. Furthermore, it's unclear why Authors did not use validated scores for MASLD/MASH to examine the liver damage.
- Evaluation of Foxo1 protein levels is an important step forward. However, reduced Foxo1 levels should favour insulin sensitivity and reduce liver damage. Results of the last experiment do not seem to be consistent.
- I still believe it is important to evaluate the translational relevance of the study findings by assessing public resources (datasets of human MASLD), or this must be discussed as a major limitation, leaving open the question as to whether the results are relevant for human disease.

Reviewer #2 (Remarks to the Author):

Please note that using a two-tailed Student's t-test with a sample size of $n=2$ is not recommended. Consideration of alternative statistical methods or additional replicates should be explored to strengthen the statistical analysis in figures 3F, 3G, 6D, 6E, 7K, and 7L.

Point-by-Point Response to Reviewers' Comments for NCOMMS-23-22157A

Reviewer #1:

1. *Response to point 1: here Authors have accumulated several different concerns, but answered only to the last one! Please address the others as well (at least limitations and findings that still do not fit with the Authors' hypothesis must be discussed)*

We added the following discussion to the text on p.5. "Mice deficient in IFN γ were shown to have blunted macrophage response, reduced adipose tissue inflammation, and attenuated insulin resistance after high-fat feeding as measured by insulin tolerance tests^{14,39,40}. We also found improved insulin sensitivity in global IFN γ null mice (obtained from the Jackson Laboratory) after high-fat feeding, and this was mostly due to improved whole-body glucose turnover and glucose metabolism in skeletal muscle (Supp. Fig. 1A & B). However, liver glucose metabolism and insulin action were not significantly affected by the systemic loss of IFN γ (Supp. Fig. 1C). The organ-selective effects of IFN γ on insulin sensitivity may be due to the pleiotropic effects of IFN γ . While IFN γ was shown to induce adipose tissue inflammation and insulin resistance in peripheral organs, IFN γ also regulates neuronal connectivity⁴¹⁻⁴³. Thus, a systemic loss of IFN γ may affect multiple organs and different cellular processes leading to potentially adaptive response, limiting the interpretation of findings from global IFN γ null mice. Importantly, these results indicate that IFN γ per se may not directly affect hepatic glucose metabolism and insulin action in obesity."

2. *Hydroxyproline content measurement is still considered the gold standard for fibrosis assessment; it's true there may be some sampling variability, but this is also applies and is even more marked for liver histology, especially given the differences in staining intensity in the "representative samples" shown. Fibrosis cannot be assessed by ultrasonography. Furthermore, it's unclear why Authors did not use validated scores for MASLD/MASH to examine the liver damage.*

We have now performed the hydroxyproline content measurement in the liver samples, and the new data, showing 33 to 43% decreases in Lyz-IFN γ R2 KO mice compared to WT mice following 20 weeks on the MASH diets, are shown in Fig. 6A and Supp. Fig. 2B and discussed in the text (p.10). We have also added the MASLD Activity Score (MAS) as validated scores for MASLD/MASH, and the data are shown in Fig. 5E and discussed in the text (p.9). The MAS was only modestly lower in Lyz-IFN γ R2 KO mice compared to WT mice because the KO mice showed lower lobular inflammation and ballooning scores but not steatosis score. These histology scores are consistent with ultrasound imaging data showing that Lyz-IFN γ R2 KO mice are protected from liver fibrosis but not steatosis following MASH diets. This is also discussed in the revised text (p.9).

3. *Evaluation of Foxo1 protein levels is an important step forward. However, reduced Foxo1 levels should favour insulin sensitivity and reduce liver damage. Results of the last experiment do not seem to be consistent.*

We have performed new RT-qPCR analysis in hepatocytes isolated from the livers of WT mice following a 4-hour infusion of IL-12 or saline (n=4 for each group). Our new data indicate that hepatocyte expression of *FoxO1* was decreased by 41% following IL-12 treatment (Fig. 7E), consistent with the liver *FoxO1* protein data (Fig. 7C). We have also added the following discussion to the revised text (p.10): “This may be due to IL-12 activation of p38-MAPK, which was shown to regulate the transcriptional activity of *FoxO1* via phosphorylation⁷⁶. Additionally, Deng et al. have found that pro-inflammatory cytokines, IL-2, IL-12, and IL-15, induce phosphorylation of *FoxO1*, leading to its inactivation in NK cells⁷⁷. IL-12 was also shown to promote the highest level of *FoxO1* phosphorylation after 2 hours of treatment⁷⁷. Thus, hepatocyte transcription factor *FoxO1* may be competitively regulated by insulin and pro-inflammatory cytokines, possibly leading to different pathogenic outcomes for the liver^{49,78}.” Moreover, a recent study from Pan et al. found that CCL4-induced liver fibrosis was associated with increased *FoxO1* protein levels in hepatocytes, and paradoxically, CCL4 also activated the hepatic Akt signaling pathway, which would stimulate *FoxO1* degradation. They reasoned that this paradox may be due to cAMP activation overriding the effect of Akt signaling in control of *FoxO1* levels as constitutive phosphorylation of *FoxO1* blocked the insulin-PI3K-Akt signaling induced *FoxO1* degradation (Pan, Q. et al., *Cell. Mol. Gastroenterol. Hepatol.*, 2024, 17:41-58; Wu, Y. et al., *Diabetes*, 2018, 67:2167-2182). These findings collectively indicate a complex regulation of liver *FoxO1* pathway.

4. *I still believe it is important to evaluate the translational relevance of the study findings by assessing public resources (datasets of human MASLD), or this must be discussed as a major limitation, leaving open the question as to whether the results are relevant for human disease.*

We analyzed RNA-seq datasets from liver samples obtained from 206 patients diagnosed with MASLD and MASH using the NICBI GEO database and found that liver expression of *IFN γ R1*, *IFN γ R2*, *IL-12B*, and *FGF21* were significantly elevated in human subjects with advancing stages of fibrosis (F2 to F4 MASH). These results signify the translational relevance of our findings to human MASH. The new data are shown in Fig. 10 and discussed on p.12.

Reviewer #2:

1. *Please note that using a two-tailed Student's t-test with a sample size of n=2 is not recommended. Consideration of alternative statistical methods or additional replicates should be explored to strengthen the statistical analysis in figures 3F, 3G, 6D, 6E, 7K, and 7L.*

We have performed new experiments using the Jess Multiplexed Western Blot System in liver samples from IL-12 and saline (controls) treated mice (n=3 for each group) for total STAT4 and phospho-STAT4 protein levels. The new data are shown in Fig. 3F and discussed in the text (p.8). We have also performed new RT-qPCR analysis in hepatocytes isolated from the livers of WT mice following a 4-hour infusion of IL-12 or saline (controls) (n=4 for each group) for *STAT4*, *p38-MAPK*, *IRS-1*, and *Akt* genes. The new data are shown in Fig. 3I and 3J and discussed in the text (p.8). For these data, one-

tailed t-test was used for statistical analysis with the rationale that STAT4 is a known activator of IL-12 signaling in the cells. We have also performed new RT-qPCR analysis in hepatocytes isolated from the livers of WT mice following a 4-hour infusion of IL-12 or saline (controls) (n=4 for each group) for *IRS-2*, *FoxO1*, and *FGF21* genes. The new data are shown in Fig. 7D, 7E, and 7F and discussed in the text (p.10). Lastly, with the assistance of Dr. Ozkan Aydemir (Director of Bioinformatics at UMass Chan Medical School), we performed a two-tailed Mann-Whitney test for statistical analysis of liver cytokine data from IL12-treated WT and Lyz-IFN γ R2 KO mice following the GAN diet as shown in Fig. 8K and 8L. The new statistical analysis is described in the figure legends.

REVIEWERS' COMMENTS

Reviewer #1 (Remarks to the Author):

My last comments were nicely addressed, well done!

Reviewer #2 (Remarks to the Author):

My comments are mostly addressed in this revised version. I have no additional comments.